# Unravelling the immune signature of *Plasmodium falciparum* transmission-reducing immunity

Will J.R. Stone[1,2], Joseph J. Campo[3], André Lin Ouédraogo[4], Lisette Meerstein-Kessel[1], Isabelle Morlais[5,6], Dari Da[7], Anna Cohuet [6,7], Sandrine Nsango[5,8], Colin J. Sutherland[2], Marga van de Vegte-Bolmer[9], Rianne Siebelink-Stoter[9], Geert-Jan van Gemert[9], Wouter Graumans[9], Kjerstin Lanke[9], Adam D. Shandling[3], Jozelyn V. Pablo[3], Andy A. Teng[3], Sophie Jones[2], Roos M. de Jong[9], Amanda Fabra-García[9], John Bradley[10], Will Roeffen[9], Edwin Lasonder[11], Giuliana Gremo[12], Evelin Schwarzer [12], Chris J. Janse[13], Susheel K. Singh[14,15], Michael Theisen[14,15], Phil Felgner[16], Matthias Marti[17,18], Chris Drakeley[2], Robert Sauerwein[1], Teun Bousema[1,2] & Matthijs M. Jore[9]

Infection with *Plasmodium* can elicit antibodies that inhibit parasite survival in the mosquito, when they are ingested in an infectious blood meal. Here, we determine the transmission-reducing activity (TRA) of naturally acquired antibodies from 648 malaria-exposed individuals using lab-based mosquito-feeding assays. Transmission inhibition is significantly associated with antibody responses to Pfs48/45, Pfs230, and to 43 novel gametocyte proteins assessed by protein microarray. In field-based mosquito-feeding assays the likelihood and rate of mosquito infection are significantly lower for individuals reactive to Pfs48/45, Pfs230 or to combinations of the novel TRA-associated proteins. We also show that naturally acquired purified antibodies against key transmission-blocking epitopes of Pfs48/45 and Pfs230 are mechanistically involved in TRA, whereas sera depleted of these antibodies retain high-level, complement-independent TRA. Our analysis demonstrates that host antibody responses to gametocyte proteins are associated with reduced malaria transmission efficiency from humans to mosquitoes.

[1] Radboud Institute for Health Sciences, Radboud University Medical Center, PO Box 9101, 6500 HB Nijmegen, The Netherlands. [2] Department of Immunology and Infection, London School of Hygiene and Tropical Medicine, Keppel Street, London WC1E 7HT, UK. [3] Antigen Discovery Inc., Irvine CA 92618, USA. [4] Institute for Disease Modeling, 3150 139th Ave SE, Bellevue, WA 98005, USA. [5] Organisation de Coordination pour la lutte contre les Endémies en Afrique Centrale, BP 288 Yaoundé, Cameroon. [6] Institut de Recherche pour le Développement, MIVEGEC (IRD, CNRS, Univ. Montpellier), 911 Avenue Agropolis, 34394 Montpellier, France. [7] Institut de Recherche en Sciences de la Santé, 399 Avenue de la Liberté, 01 BP 545 Bobo-Dioulasso, Burkina Faso. [8] Faculty of Medecine and Pharmaceutical Science, PO Box 2701, Douala, Cameroon. [9] Radboud Institute for Molecular Life Sciences, Radboud University Medical Center, PO Box 9101, 6500 HB Nijmegen, The Netherlands. [10] Medical Research Council Tropical Epidemiology Group, London School of Hygiene and Tropical Medicine, Keppel Street, London WC1E 7HT, UK. [11] School of Biomedical and Healthcare Sciences, Plymouth University, Drakes Circus, Plymouth PL4 8AA, UK. [12] Department of Oncology, University of Torino, Via Santena 5bis, 10126 Torino, Italy. [13] Department of Parasitology, Leiden University Medical Center (LUMC), Albinusdreef 2, 2333 ZA Leiden, The Netherlands. [14] Department for Congenital Diseases, Statens Serum Institut, Copenhagen DK 2300, Denmark. [15] Centre for Medical Parasitology at Department of International Health, Immunology and Microbiology, University of Copenhagen and Department of Infectious Diseases, Copenhagen University Hospital, Rigshospitalet, Copenhagen DK 2200, Denmark. [16] Department of Medicine, University of California Irvine, Irvine CA 92697, USA. [17] Department of Immunology and Infectious Diseases, Harvard School of Public Health, Boston MA 02115, USA. [18] Wellcome Center for Molecular Parasitology, University of Glasgow, Glasgow G12 8TA, UK. Teun Bousema and Matthijs M. Jore jointly supervised this work. Correspondence and requests for materials should be addressed to W.J.R.S. (email: william.stone@lshtm.ac.uk) or to T.B. (email: teun.bousema@radboudumc.nl)

Plasmodium gametocytes develop from their asexual progenitors and are the only malaria parasite life-stage infective to mosquitoes. In preparation for their development in the mosquito, gametocytes cease to express many proteins involved in the parasites cycle of asexual replication, and upregulate others that are involved in sexual development[1,2]. Some of these have essential roles in mosquito-stage development[3,4], which culminates in the insect becoming infectious to humans.

In surveys where mosquitoes were fed directly on the skin or on a blood sample, it was noted that some gametocytaemic individuals including those with high gametocyte densities were not infectious[5,6]. This may be associated with naturally acquired antibodies that interfere with parasite development within mosquitoes to reduce or prevent mosquito infection[7,8]. The role of antibodies in determining transmission efficiency can be tested in mosquito-feeding assays, in which purified antibodies are added to a mosquito blood meal that contains gametocytes[9,10]. Evidence for naturally acquired transmission-reducing activity (TRA) is provided if test antibodies in endemic serum cause a reduction in the number of developing Plasmodium oocysts relative to mosquitoes fed the same infectious blood meal without test antibodies. TRA can be experimentally induced after immunisation of animals with inactivated Plasmodium gametocytes or gametes[8,11]. The gametocyte and gamete proteins P48/45 and P230 have been identified as targets of transmission-blocking monoclonal antibodies (mAb), which act by inhibiting the protein's role in gamete fertilisation in the mosquito gut[3,4,12]. Monoclonal antibodies specific to the zygote and ookinete proteins P25 and P28 were also shown to block transmission, by inhibiting ookinete invasion of the midgut epithelium[13]. All four have been produced as recombinant proteins that can induce antibody-mediated TRA in animal models[13–15].

Because P25 and P28 mRNA is translationally repressed until zygote formation[16], antibodies specific to these proteins appear absent in endemic populations[17,18]. In contrast, responses to P48/45 and P230 are commonly observed in naturally infected individuals[10,19,20]. The presence of antibodies to Plasmodium falciparum Pfs48/45 and Pfs230 in endemic sera has been associated with TRA in several, but not all sero-epidemiological surveys[10,21–28]. Importantly, many individuals with functional TRA do not have measurable antibodies against these two proteins[10,23,25,28–31]. Despite these observations, there has been no attempt to demonstrate the contribution of naturally acquired α-Pfs48/45 and α-Pfs230 antibodies to transmission inhibition, and little investigation of alternative targets of natural TRA. The gametocyte proteome has now been described in detail[32,33]. Utilising the protein microarray platform, genome scale data sets have been combined with functional and immunological data to provide valuable insight into mechanisms and markers of malaria humoral immunity[34,35].

Here, we aim to investigate the immune signature of naturally occurring antibody-mediated TRA, to expand and prioritise antigenic targets for functional characterisation as biomarkers, or transmission-blocking vaccine candidates. To this purpose, we utilise a protein microarray comprising an inclusive selection of proteins expressed during gametocyte development. We assess antibody responses to these proteins and to correctly folded Pfs48/45[36] and Pfs230[15] in 648 malaria-exposed individuals from Burkina Faso, the Gambia, Cameroon, and from migrants from the Netherlands. Using purified total IgG, we assess the functional TRA of antibodies from the same individuals by determining their effect on mosquito infection density in standard membrane-feeding assays (SMFA) with cultured P. falciparum gametocytes and Anopheles stephensi mosquitoes. Our analysis reveals significant associations between high-level TRA and responses to Pfs48/45, Pfs230 and 43 novel gametocyte proteins. For a subset of 366 gametocyte-positive donors who had provided blood samples to Anopheles gambiae s.s. or An. coluzzii mosquitoes in field-based membrane-feeding assays, we determine the association of these antibody responses with mosquito infection rates during natural infections. For Pfs48/45 and Pfs230, we provide functional data that demonstrate their role in natural TRA. For one of the newly identified proteins (PfGEST), we provide experimental data that does not support its functional role in natural TRA.

## Results

**Antibody responses to gametocyte antigens**. Plasma was collected in epidemiological studies in Burkina Faso[37–41], Cameroon[42,43] and the Gambia[24,44–48], as well as from Dutch migrants who had lived for several years in malaria-endemic areas and reported repeated malaria episodes (Table 1). Individuals from malaria-endemic areas were all asymptomatic when sampled, and were either recruited randomly from the community (n = 42), or based on the observation of malaria parasites (n = 273) or specifically gametocytes (n = 276) in blood smears. For a selection of individuals from malaria-endemic areas (n = 498), infectivity to mosquitoes at the moment of sample collection was determined by offering a venous blood sample to Anopheles gambiae s.s. or An. coluzzii in a direct membrane-feeding assay (DMFA) (Table 1). Dutch migrants had no DMFA performed and had either recently returned from extended periods in endemic areas (within 6 months after return; n = 8) or had returned more than a year before sampling (n = 49). Those who had returned more than a year before sampling had lived in areas with endemic P. falciparum transmission for longer than 15 years,

---

**Table 1 Sample characteristics**

| Sample origin (study reference) | Samples (n) | Age (median, range) | Asexual positive (%) | Gametocyte-positive (%) | Parasite free (%) | Infectious (DMFA, %) | TR activity | | | |
|---|---|---|---|---|---|---|---|---|---|---|
| | | | | | | | <10% | ≥50% | ≥80% | ≥90% |
| Gambia[24, 44–48] | 226 | 5.0 (1.0–17.0) | 99.1% (224/226) | 74.2% (167/225) | 0.4% (1/226) | 24.0% (44/183) | 54.0% (122/226) | 15.9% (36/226) | 4.4% (10/226) | 3.1% (7/226) |
| Burkina Faso[24, 44–48] | 225 | 16.0 (1.8–74.0) | 56.5% (109/193) | 53.5% (114/213) | 31.9% (68/213) | 45.5% (86/189) | 68.4% (154/225) | 8.0% (18/225) | 2.7% (6/225) | 1.8% (4/225) |
| Cameroon[42, 43] | 140 | 8.5 (5.0–16.0) | 90.5% (117/133) | 93.5% (137/138) | 0.0% (0/139) | 73.8% (93/126) | 68.6% (96/140) | 14.3% (20/140) | 6.4% (9/140) | 4.3% (6/140) |
| The Netherlands[37–39, 43] | 57 | 79.5 (26.0–84.0) | 3.9% (2/51) | 3.9% (2/51) | 96.1% (49/51) | — | 57.9% (33/57) | 17.5% (10/57) | 8.8% (5/57) | 8.8% (5/57) |
| Combined | 649 | 7.1 (1–92) | 75.6% (458/606) | 65.8% (413/628) | 18.8% (118/629) | 44.8% (223/498) | 62.5% (405/648) | 13.0% (84/648) | 4.6% (30/648) | 3.4% (22/649) |

"—" data unavailable, or untested; *TR activity (%)*: percent transmission-reducing (TR) activity of purified IgG in the standard membrane-feeding assay (SMFA), relative to control mosquitoes fed the same gametocyte batch without test antibodies. TR activity is the mean of two independent SMFA runs for all samples with ≥80% TR activity in the first run, *Samples*: total number of samples with TR activity assessed in the SMFA, *Asex/Gct %*: asexual parasite and gametocyte prevalence by microscopy at the time of sampling, for individuals with available data, *Parasite free %*: no asexual parasites or gametocytes observed by microscopy

---

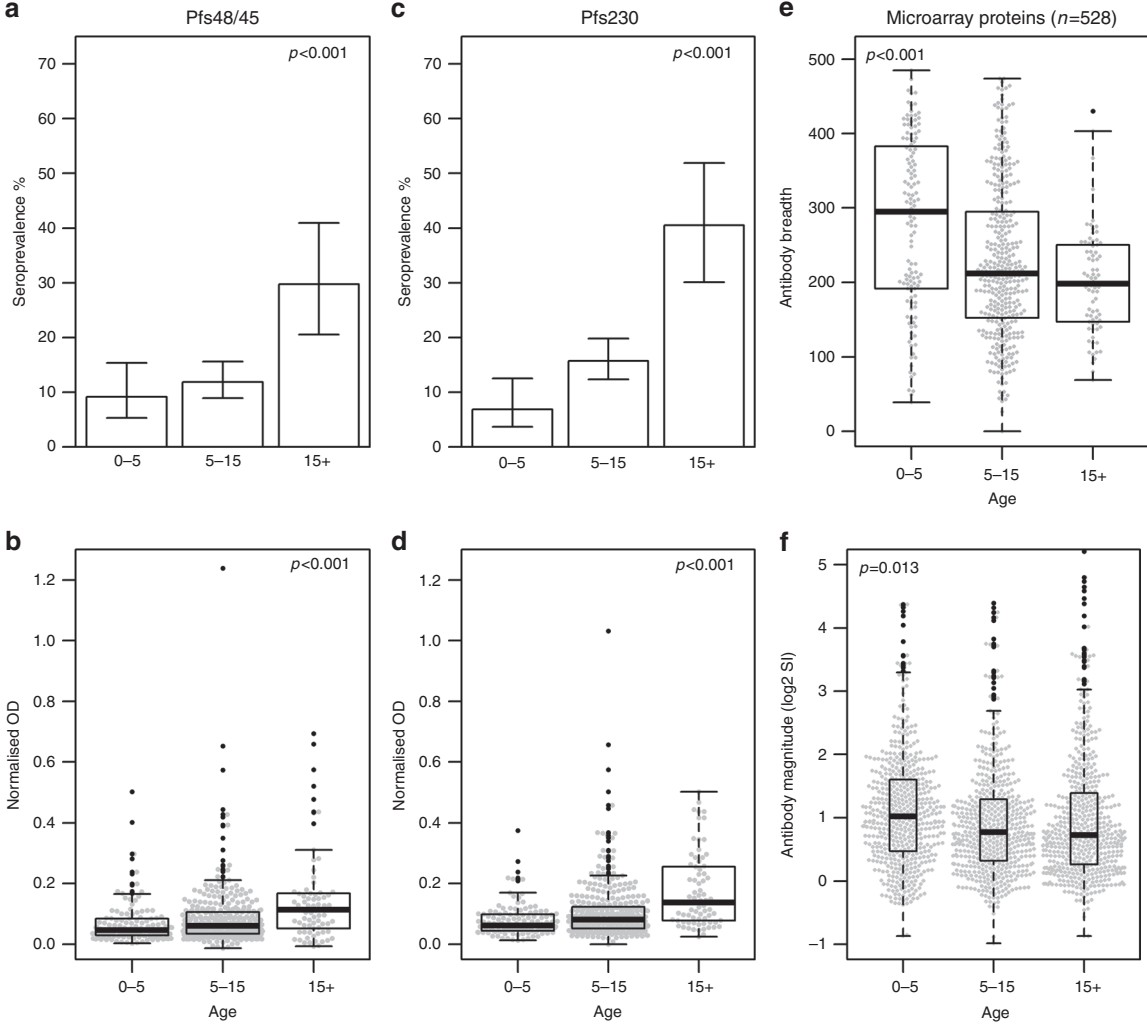

**Fig. 1** Antibody responses to Pfs48/45, Pfs230 and to the microarray proteins with age. Antibody responses to correctly folded, recombinant Pfs48/45-10C and Pfs230 (230CMB) were measured using ELISA, with antibody intensity given as the ELISA optical density (OD) values (450 nm). Antibody responses to microarray proteins are given as the $\log_2$-transformed signal intensity (SI) minus the vehicle SI, which equates to the $\log_2$-fold change over this background. All graphs show only individuals from endemic areas (Dutch migrants excluded). Sample size: 0–5 = 131, 5–15 = 366, 15+ = 71. **a**, **c** Bars show the seroprevalence of α-Pfs48/45 and α-Pfs230 antibodies with age, with Clopper–Pearson confidence intervals. **b**, **d** Boxplots showing α-Pfs48/45 and α-Pfs230 antibody intensity with age. **e** Boxplots showing responses to the microarray protein targets (n = 528). Antibody breadth is the number of proteins reactive above background in each individual, within the given groups. **f** Magnitude of antibody response to microarray protein targets. Each dot represents the average SI of response to each protein target by all individuals within given groups. p-values for prevalence data are from likelihood ratio test for differences in seroprevalence between all age groups, derived from logistic regression and adjusted for gametocyte density. p-values for intensity data and response breadth are from an F-test for differences in OD/SI between all age groups, derived from linear regression and adjusted for gametocyte density or from ANOVA (for magnitude only). For all boxplots, outliers are shown in black, whereas all data points are shown in grey as a bee-swarm

and reported between 2 and >80 clinical malaria episodes during that time.

Antibody responses to Pfs48/45 (Pfs48/45-10C[36]) and Pfs230 (230CMB[15]) were determined by enzyme-linked immunosorbent assay (ELISA) using recombinant, correctly folded proteins. A protein microarray was constructed to assess responses to other gametocyte antigens. This array included a total of 315 proteins (Supplementary Data 1) that were broken up into 528 overlapping fragments <1000 amino acids in length (overlaps of 17 amino acids) for protein expression in an *Escherichia coli* based in vitro transcription–translation (IVTT) system[49,50]. Antigens on the array included proteins previously implicated as eliciting TRA after immunisation, proteins involved in early gametocyte development as possible markers of gametocyte exposure[1], markers of asexual parasite exposure[51] and proteins expressed

in gametocytes based on recent proteomic analysis[52]. To ensure that the array included most potential antibody targets involved in TRA, we preferentially included proteins that had transmembrane domains, signal peptides or GPI anchors. Specificity of proteins to gametocytes was not a prerequisite for inclusion on the array, however 228/315 proteins had consensus evidence for enrichment in gametocytes based on a recent integration analysis of 11 available proteomics data sets[53] (Supplementary Data 1). Among donors from malaria-endemic regions, the prevalence and magnitude of antibody responses to Pfs48/45 and Pfs230 increased with age (Fig. 1a–d). In contrast, the overall breadth (linear regression (LinR): $p < 0.001$) and magnitude (analysis of variance (ANOVA): $p = 0.013$) of response to antigens on the microarray decreased with age (Fig. 1e, f). The same associations were observed when analyses were restricted to donors from

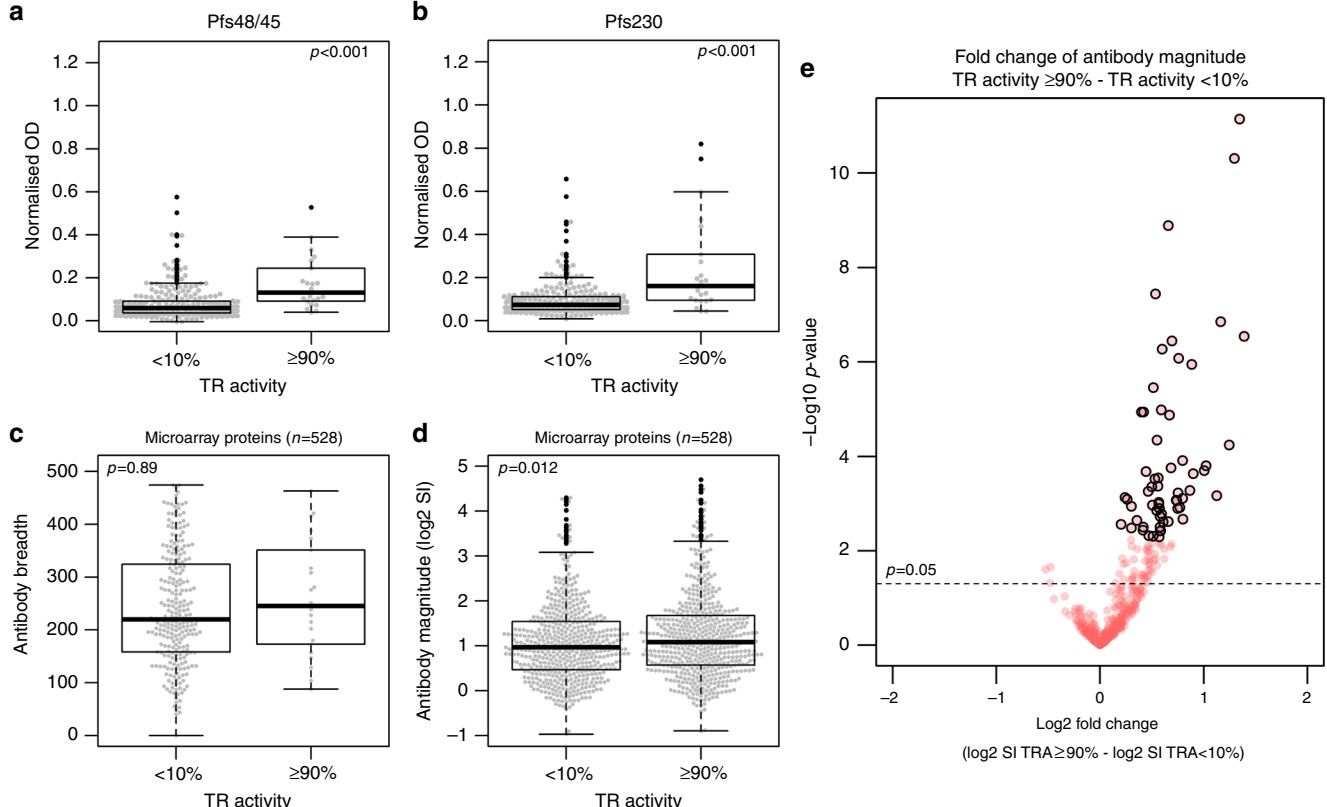

**Fig. 2** Transmission-reducing activity (TRA) and antibody responses to Pfs48/45, Pfs230 and microarray proteins. TRA was categorised as described, to compare responses between gametocyte-positive individuals with <10% TRA, and individuals with >90% TRA. Antibody responses to correctly folded, recombinant Pfs48/45-10C and Pfs230 (230CMB) were measured using ELISA, with antibody intensity given as the ELISA optical density (OD) values (450 nm). Antibody responses to microarray proteins are given as the $\log_2$-transformed signal intensity (SI) minus the vehicle SI, which equates to the $\log_2$-fold change over this background. **a**, **b** Boxplots of α-Pfs48/45 and α-Pfs230 antibody intensity with TRA. **c** Responses to the microarray protein targets ($n = 528$). Antibody breadth is the number of proteins reactive above background in each individual, within the given groups. **d** Magnitude of antibody response to microarray protein targets. Each dot represents the average SI of response to each protein target by all individuals within given groups. $p$-values for intensity data and response breadth are from an $F$-test for differences in OD/SI between all age groups, derived from linear regression, and adjusted for gametocyte density or from students $t$-test (for magnitude only). For all boxplots, outliers are shown in black, whereas all data points are shown in grey as a bee-swarm. **e** Volcano plot showing the $\log_2$-fold change of the mean signal intensity for TRA as defined above. The $p$-value shown by the dotted line is unadjusted for false discovery. Black-circled data points are those that remain significant after $p$-values have been adjusted to control the rate of false discovery below 5%, using the Benjamini–Hochberg method

Burkina Faso ($n = 225$), the site with the widest age range of donors (Supplementary Fig. 1).

**Antibody responses and transmission-reducing activity (TRA).** To assess functional TRA, IgG was purified from the sera of all 648 individuals and provided to *Anopheles stephensi* mosquitoes together with a blood meal containing cultured NF54 or transgenic NF54HT-GFP-Luc gametocytes in standard membrane-feeding assays (SMFA)[9,54]. If the donor IgG caused ≥80% reduction in oocyst formation relative to control mosquitoes (TRA ≥80%), the SMFA was repeated with separate gametocyte culture material. In total, 3.3% (22/648) of the antibody samples caused replicable TRA of ≥90% (Table 1). On a continuous scale, TRA decreased with age, but was not significantly associated with gametocyte density at the point of sampling (LinR: $p = 0.269$) (Supplementary Fig. 2). Because previous gametocyte exposure is associated with TRA[28], comparisons between sub-groups were nevertheless adjusted for gametocyte density. Though intermediate TRA may be relevant for transmission there are uncertainties about its reliable measurement in the SMFA[55,56]. TRA was therefore dichotomised into ≥90% (high-level TRA) and

<10% TRA (no TRA) before assessing associations with antibody responses. The group without evidence for TRA was refined by excluding individuals without microscopically detectable gametocyte densities at the time of sampling, resulting in 22 individuals with high-level TRA activity and 254 individuals who were exposed to gametocytes (and thus likely to respond to gametocyte antigens) but showed no evidence for TRA. Using these two categories, high-level TRA was significantly more likely in individuals seropositive for α-Pfs230 (odds-ratio (OR) from logistic regression (LogR) 7.9 (95% confidence interval (CI): 2.8–22.8), $p$ < 0.001), α-Pfs48/45 (OR 4.4 (95% CI: 1.5–12.9), $p = 0.007$) or either antigen (OR 5.90 (95% CI: 2.1–16.7), $p = 0.001$) (Supplementary Table 1, Supplementary Data 2). Antibody density was also higher for Pfs230 (LinR: $p < 0.001$) and Pfs48/45 ($p < 0.001$) in individuals exhibiting high-level TRA (Fig. 2a, b; Supplementary Data 2).

In addition to Pfs230 and Pfs48/45, the average magnitude of response to each microarray target was also higher in individuals with high-level TRA (students $t$-test: $p = 0.012$) (Fig. 2d), whilst the breadth of response by each individual was not statistically significantly different between individuals with high-level TRA

compared to those with no evidence of TRA (LinR: $p = 0.89$) (Fig. 2c). Sixty microarray proteins had statistically significantly higher antibody magnitude in individuals with high-level TRA, after $p$-values had been adjusted to control the false discovery rate (FDR) (Fig. 2e)[57]. The inverse, higher responses in individuals with no TRA, was not observed (Supplementary Data 3). Logistic regression models with FDR controlled $p$-values showed that antibody prevalence to 27 microarray targets was significantly associated with high-level TRA, 25 of which were also significantly more reactive in the analysis of signal intensity. Because large proteins were fragmented on the array, the combined list of 62 novel microarray targets with TRA-associated antibody responses in either the linear (response intensity) or logistic (response prevalence) analysis represented fragments of 43 unique proteins (Supplementary Data 2). 15 of the 43 proteins with TRA-associated responses are conserved unknown proteins. Two are known gametocyte/gamete proteins with putative roles in gamete egress; PF3D7_1038400 (gametocyte-specific protein, Pf11-1) and PF3D7_1449000 (gamete surface and sporozoite traversal protein/GEST).

TRA-associated responses on the array showed only partial overlap with Pfs48/45 and Pfs230 responses. Individuals with high-level TRA who were seronegative for both α-Pfs230 and α-Pfs48/45 ($n = 10/22$), had antibody responses to significantly more TRA-associated microarray targets (median breadth 11 (out of 62) (interquartile range (IQR) 3–26)) than gametocyte-positive individuals without TRA ($n = 211$; median breadth 4 (out of 62) (IQR 2–9), LinR: $p = 0.018$). Responses to some TRA biomarkers were highly prevalent in individuals with high-level TRA who lacked significant responses to Pfs45/45 and Pfs230 (e.g. Pf11-1 (PF3D7_103840) = 70% ($n/N = 7/10$); LSA3 (PF3D7_0220000) = 80% ($n/N = 8/10$)). Responses to one target protein mapping to PF3D7_1103800 (CCR4-NOT) were present in 60% ($n/N = 6/10$) of high-level TRA α-Pfs48/45 and α-Pfs230 seronegatives, whereas seroprevalence for this target in individuals with no TRA was 10.3% ($n/N = 26/254$) (Supplementary Data 4).

Interestingly, 10 of the 43 proteins with TRA-associated antibody responses are putatively exported during early gametogenesis[1]. Though the majority of these proteins are enriched in asexual parasites, their association with TRA may reflect previous observations that TR immunity is induced and wanes rapidly after gametocyte exposure[26,27]. Responses to putative markers of asexual parasite exposure (PF3D7_1036000 (Merozoite surface protein 11 (MSP11)), PF3D7_0711700 (erythrocyte membrane protein 1, PfEMP1 (VAR)), PF3D7_0731600 (acyl-CoA synthetase (ACS5)), and PF3D7_0423700 (early-transcribed membrane protein 4 (ETRAMP4))[51] were not associated with TRA, either in terms of response intensity (Bayes empirical $t$-test: FDR-adjusted $p = 0.34$–0.99) or seroprevalence (LogR: FDR-adjusted $p = 1$).

**TRA and mosquito infection rate in natural infections**. Next, we determined whether the immune profile associated with TRA in the SMFA was also associated with mosquito infection rates observed in direct membrane-feeding experiments. Data on the infectivity of donor blood samples to mosquitoes at the time of plasma donation was available for 494 individuals for whom gametocyte status had been assessed by microscopy and TRA assessed in the SMFA. Of the 128 individuals without patent gametocytaemia, 18 (14.1%) were infective to mosquitoes in the DMFA. Of the 366 gametocyte-positive individuals, 203 (55.5%) infected at least one mosquito, with a median mosquito infection rate of 20% (IQR 7.7–42.5%). There was good agreement between transmission outcomes in the SMFA using cultured gametocytes and *An. stephensi* mosquitoes, and the field-based DMFA using gametocytes in infected individuals and *An. gambiae s.s.* or *An.*

*coluzzii* mosquitoes; Individuals with high-level TRA in the SMFA were significantly less likely to infect any mosquitoes in the DMFA (LogR: OR 0.23 (95% CI: 0.06–0.79), $p = 0.020$), and more likely to do so at a reduced rate (OR 0.10 (95% CI: 0.2–0.61), $p = 0.012$). The likelihood of gametocyte-positive individuals infecting any mosquitoes in the DMFA was significantly lower if seropositive for Pfs230 (LogR: OR 0.42 (95% CI: 0.22–0.78), $p = 0.006$), Pfs48/45 (OR 0.30 (95% CI: 0.16–0.59), $p < 0.001$), or either or both antigens (OR 0.33 (95% CI: 0.19–0.58), $p < 0.001$) (Fig. 3a). The proportion of mosquitoes that became infected after feeding was also significantly lower for individuals seropositive for Pfs230 (LogR: OR 0.36 (95% CI: 0.14–0.88), $p < 0.027$), Pfs48/45 (OR 0.14 (95% CI: 0.05–0.40), $p < 0.001$) or either or both antigens (OR 0.25 (95% CI: 0.11–0.55), $p = 0.001$) (Fig. 3b). The proportion of mosquitoes that became infected decreased with increasing Pfs230 antibody density (LogR: OR for an increase of 0.1 normalised optical density 0.59 (95% CI: 0.40–0.86), $p = 0.007$) and Pfs48/45 antibody density (OR 0.43 (95% CI: 0.28–0.66), $p < 0.001$).

Mosquito infection prevalence was lower in gametocyte-positive individuals with antibodies recognising more of the 62 novel protein microarray targets with TRA-associated antibody responses (Supplementary Fig. 3). Individuals responding to ≥14/62 microarray proteins (14 being the 75th percentile of breadth of response to these targets) were significantly less likely to infect mosquitoes (LogR: OR 0.31 (95% CI: 0.18–0.51), $p < 0.001$), and more likely to do so at a reduced rate (OR 0.21 (95% CI: 0.09–0.45). $p < 0.001$) (Fig. 3a, b). After excluding Pfs48/45 or Pfs230 seropositive individuals, infectiousness and infection rate remained significantly lower in individuals responding to ≥14 of the 62 TRA-associated microarray proteins (infectiousness: LogR, OR 0.29 (95% CI: 0.16–0.53), $p < 0.001$, infection rate: OR 0.23 (95% CI: 0.10–0.52), $p < 0.001$).

**Functional involvement of antibodies in TRA**. To demonstrate whether naturally acquired antibodies to Pfs230 or Pfs48/45 were causally associated with TRA, we affinity-purified antibodies specific to key transmission-blocking epitopes of these proteins and assessed their activity in the SMFA (Table 2). These experiments were performed for 6 donors whose IgG showed high-level TRA and for whom high volumes of plasma (≥1 mL) were available; 3 of these donors had ≥10 mL plasma available, allowing additional testing after 9-fold antibody concentration of IgG. The flow-through of affinity purification experiments, depleted of α-Pfs230 and α-Pfs48/45 antibodies, was also tested in the SMFA to quantify TRA of antibodies to other target antigens. Concentrated back to the original plasma volume, α-Pfs48/45 antibodies of one donor independently inhibited transmission (TRA 91.5% (95% CI: 86.4–94.7)), whereas α-Pfs230 antibodies of a different donor had low but statistically significant TRA (TRA 41.1% (95% CI: 15.5–60.4)). When antibodies were concentrated to nine times their physiological concentration, α-Pfs48/45 antibodies from one additional individual, and α-Pfs230 antibodies from two additional individuals significantly reduced transmission to mosquitoes (Pfs48/45: TRA 81.3%, (95% CI: 70.9–88.0), Pfs230: TRA 94.8% (95% CI: 90.2–97.3) & 99.3% (95% 98.6–99.6), Table 2). For all individuals the α-Pfs48/45 and α-Pfs230-depleted IgG (containing no antibodies against Pfs48-45 epitopes 1–3, or Pfs230 epitope C, confirmed by ELISA) inhibited transmission to mosquitoes (Table 2). This activity was complement-independent.

To determine the presence of naturally acquired antibodies binding to unknown gamete surface antigens, we performed surface immuno-fluorescence assays (SIFA) using gametes of a Pfs48/45 knockout (KO) parasite[3] that does not produce surface

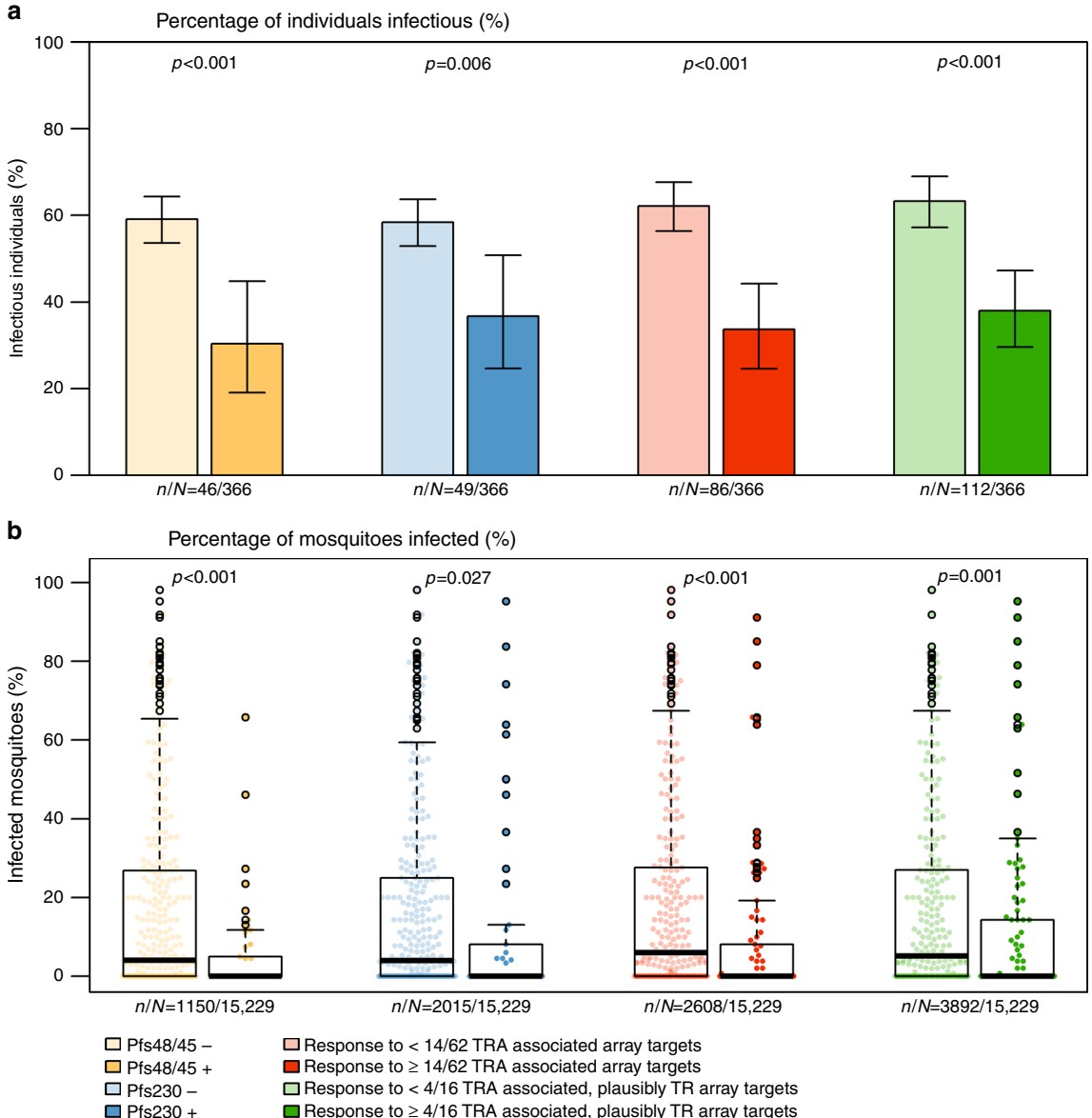

**Fig. 3** Seroprevalence to Pfs48/45, Pfs230, and novel TRA-associated microarray proteins, and infectiousness in the direct membrane-feeding assay (DMFA). Individuals with DMFA data were categorised according to their possession of antibodies specific to: Pfs48/45 (positive (+)/negative (−)); Pfs230 (positive (+)/negative (−)); ≥14 of the 61 novel microarray proteins with TRA-associated antibody responses (14 being the 75th percentile of the breadth of response to these microarray targets among the entire sample set); ≥4 of the 16 novel microarray proteins with TRA-associated antibody responses that are also plausible targets of antibodies with TRA (4 being the 75th percentile of the breadth of response to these microarray targets among the entire sample set). **a** Bars show the proportion of infectious individuals among seropositive/seronegative gametocytaemic individuals with DMFA data, with Clopper–Pearson confidence intervals. $n/N$ = number of individuals seropositive/total number of individuals with DMFA data. P-values are from logistic regression, with adjustment for gametocyte density. **b** Boxplots show the percentage of mosquitoes that became infected after feeding. $n/N$ = the number of mosquitoes feeding on seropositive individuals/the total number of mosquitoes feeding on individuals with DMFA data. p-values are from logistic models, adjusted for gametocyte density and with host (individual the mosquitoes were feeding upon) as a random effect. For all boxplots, outliers are shown as hollow black circles, whereas all data points are shown as solid coloured circles

retained Pfs230[12]. Pfs48/45- and Pfs230-specific antibodies confirmed the absence of these proteins on Pfs48/45 KO gamete surfaces (Fig. 4). The immuno-fluorescence of gamete surfaces observed using total IgG and IgG depleted of α-Pfs48/45 and α-Pfs230 antibodies from naturally exposed individuals therefore reflects recognition of unknown antigens on the gamete surface (Fig. 4; Supplementary Fig. 5A).

Antibody responses to a number of the 43 novel proteins that are associated with TRA (Supplementary Data 2) may be biomarkers of high prior gametocyte exposure rather than

causally related to TRA. We thus curated this list of proteins by excluding genes with known or predicted intracellular products, while retaining genes based on evidence for a role in transmission reduction or gamete viability, or predicted compatibility with gametocyte/gamete surface expression. In addition to Pfs230 and Pfs48/45, 13 novel proteins possess sequence characteristics that are compatible with surface expression or have known roles in gamete viability (Table 3; Supplementary Data 5) and may thus form biologically plausible targets of transmission-inhibiting antibodies. mAbs specific to Pf11-1, one of the biomarkers with

**Table 2 Activity of affinity-purified antibodies against R0.10C (Pfs48/45) and 230CMB (Pfs230) from transmission-reducing sera**

| ID | Location | Sex/age | Time since malaria exposure | TRA% (95% confidence intervals) | | | | | | | |
|----|----------|---------|-----------------------------|---------------------------------|--|--|--|--|--|--|--|
| | | | | 1. Total IgG | 2a. R0.10C IgG | 2b. R0.10C IgG (conc.) | 3. Total IgG w/o R0.10c IgG | 4a. 230CMB IgG | 4b. 230CMB IgG (conc.) | 5a. Total IgG w/o R0.10C & 230CMB IgG | 5b. Total IgG w/o R0.10C & 230CMB IgG (C−) |
| A | Netherlands | M/69 | Infected | 99.4 (98.9–99.7) | 91.5 (86.4–94.7) | ND | 99.5 (99–99.7) | −2.5* (−38.6–24.2) | 99.3 (98.6–99.6) | 99.5 (99–99.8) | 99.8 (99.5–99.9) |
| B | Netherlands | — | Infected | 99.6 (99.3–99.8) | −42.8 (−80.5–12.9) | ND | 96.7 (95.6–97.5) | −15.7* (−46.1–8.4) | ND | 88.5 (85.7–90.7) | 87 (79.9–91.6) |
| C | Netherlands | — | <6 months | 99.9 (99.7–100) | 25.6* (−4.1–46.8) | 81.3 (70.9–88) | 99.4 (99–99.7) | 42.1 (15.5–60.4) | 94.8 (90.2–97.3) | 89.1 (80.2–94) | 95.8 (91.4–98) |
| D | Netherlands | M/74 | ~5 years | 99.6 (99–99.8) | −14.7* (−65.3–20.5) | −0.3* (−38.7–27.4) | 85.4 (75.4–91.3) | 32* (−1–54.1) | 0.1* (−58.8–37.1) | 71.8 (52.7–83.2) | 80.8 (66.8–88.9) |
| E | Cameroon | F/12 | Infected | 98.8 (98–99.3) | 41.7 (26–54.1) | ND | 93.1 (90.3–95.1) | 16.5* (−8.6–35.8) | ND | 92.8 (89–95.3) | 93.4 (88.6–96.1) |
| F | Cameroon | F/6 | Infected | 100 (99.8–100) | 29.3* (−6.7–53.2) | ND | 99.9 (99.7–100) | 14.8* (−22.1–40.6) | ND | 97.2 (95.5–98.3) | 98.9 (96.7–99.6) |
| Ctrl | Netherlands | Pooled | Never | −26.5* (−72–7) | −17.2* (−50.9–9) | 4.6* (−33–31.5) | −14.7* (−50–12.3) | −6.3* (−31.4–14) | −2.7* (−41.5–25.5) | 6.5* (−19.4–26.9) | ND |

*ND*: not done. Transmission-reducing activity is from duplicate SMFA experiments, with luminescence intensity 7–9 days after mosquito-feeding (using NF54HT-GFP-luc strain gametocytes) as the output measure of infection intensity. All samples were tested in the presence of complement unless noted otherwise (C-). Total IgG represents the SMFA whole-purified patient IgG at physiological concentration. Pure IgG represents the TRA of antibodies that bound to the R0.10C and 230CMB columns. Conc. IgG represents pure antibodies at 9 times the physiological concentration. The fractions tested were as follows; 1. Total IgG, 2a. Purified α-R0.10c IgG, 2b. Concentrated (9×) α-R0.10c IgG, 3. Total IgG without α-R0.10c IgG, 4a. Purified α-230CMB IgG, 4b.Concentrated (9×) α-230CMB IgG, 5. Total IgG without α-R0.10C and α-230CMB-IgG. Fraction 5, depleted of antibodies binding to either protein column, was tested with and without (C-) complement. Ctrl is IgG from plasma from pooled naive Dutch donors. All TR values and confidence intervals were calculated from two independent SMFAs with different culture material. TRA, CI and *p*-values were calculated using generalised linear models (GLM) as described previously[72]. Asterisks (*) indicate results in which the oocyst/luminescence intensity of the test mosquitoes was not significantly different to the controls. Corrected ELISA OD values for α-10c IgG in flow-through fractions: 1. A = 0.12, B = 0.88, C = 0.15, D = 0.08, E = 0.13, F = 0.08, Ctrl = 0.02; 3. Range for all individuals 0.01–0.04. Corrected ELISA OD values for α-230CMB IgG in: 1. A = 0.05, B = 1.16, C = 0.76, D = 0.27, E = 0.10, F = 0.07, Ctrl = 0.06; 5. Range for all individuals 0.01–0.03

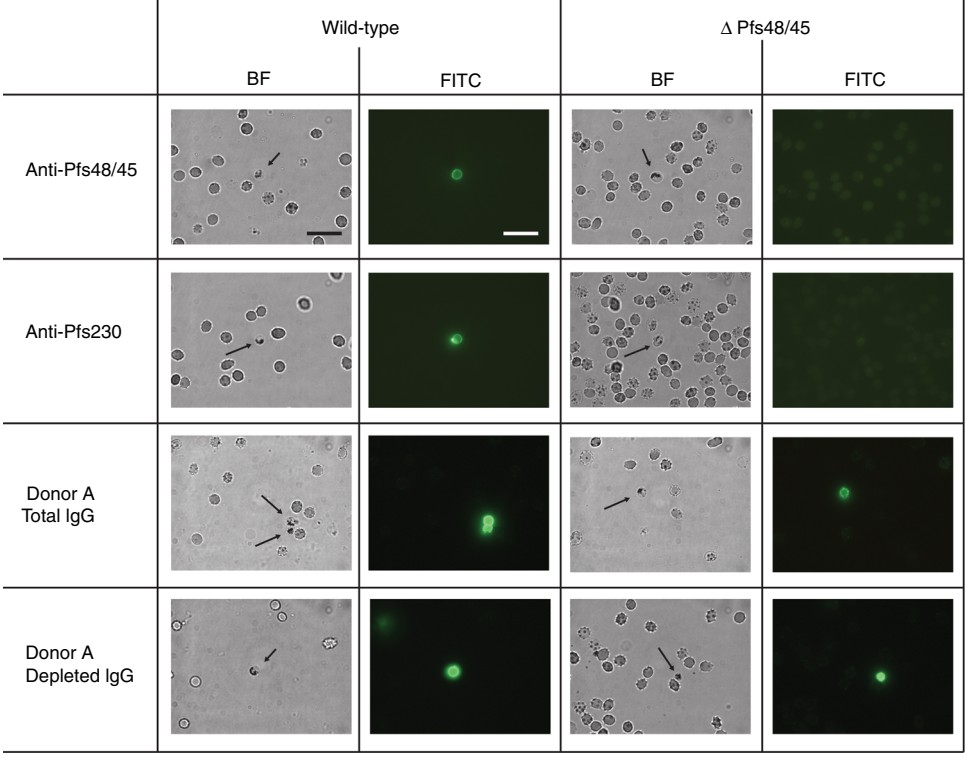

**Fig. 4** Gamete surface immuno-fluorescence assay (SIFA) using wild-type and Pfs48/45 knockout (KO) NF54 gametes, with Pfs48/45 mAb, Pfs230 mAb, and IgG from a malaria-exposed serum donor. Donor IgG is from donor A in Table 2, and was performed using total IgG, and total IgG depleted of α-Pfs48/45-10c and α-Pfs230 (230CMB) IgG. Δ Pfs48/45 = Pfs48/45 KO[3]. BF Bright-field, FITC fluorescein isothiocyanate. Anti-Pfs48/45 is mAb 45.1, and anti-Pfs230 is mAb 2A2, as described in the methods. Scale bar is 20 µm

a role in gamete egress, were shown previously to reduce transmission to mosquitoes in membrane-feeding experiments[58]. Previous data show that the *P. berghei* homologue of another of our markers, PF3D7_1449000 (PfGEST), shows partial association with gamete surfaces and has a clear role in gamete egress[59]. We tested three monoclonal antibodies (mAb) against PfGEST in the SIFA and SMFA. SIFA with PfGEST mAbs showed no, or very limited surface reactivity, indicating that

**Table 3 Proteins with TRA-associated antibody responses**

Biomarkers of TR immunity, and plausible or known gametocyte/gamete surface proteins (remaining biomarkers in Supplementary Data 5)

| Gene ID | Gene description | TM | SP | Parasite stage | Cellular component (homology/annotation) |
|---|---|---|---|---|---|
| PF3D7_0209000 | 6-cysteine protein (P230) | 0 | Y | Gametocyte-specific | Known gamete surface protein, complex with P48/45 |
| PF3D7_1346700 | 6-cysteine protein (P48/45) | 1 | Y | Gametocyte-specific | Known GPI anchored gamete surface protein |
| PF3D7_0305300 | Conserved Plasmodium membrane protein, unknown function | 11 | | Gametocyte enriched, shared | Extracellular region, membrane (transmembrane transporter) |
| PF3D7_1021100 | Conserved Plasmodium protein, unknown function | 2 | | Gametocyte-specific | Membrane |
| PF3D7_1038400 | Gametocyte-specific protein (Pf11-1) | 1 | | Gametocyte-specific | Host cell cytoplasm, implicated in gamete egress |
| PF3D7_1107900 | Mechanosensitive ion channel protein | 5 | | Gametocyte enriched, shared | Membrane |
| PF3D7_1143700 | Conserved Plasmodium protein, unknown function | 1 | | Gametocyte enriched, shared | Cytoplasm, membrane |
| PF3D7_1306500 | MORN repeat protein, putative | 10 | | Gametocyte enriched, shared | N/A (MORN, unknown cell component) |
| PF3D7_1314500 | Cop-coated vesicle membrane protein p24 precursor, putative | 2 | Y | Asexual enriched, shared | Integral component of membrane |
| PF3D7_1360500 | Guanylyl cyclase beta (GCbeta) | 21 | | Gametocyte enriched, shared | N/A (GCbeta, unknown cell component) |
| PF3D7_1433200 | Conserved Plasmodium protein, unknown function | 1 | | Gametocyte enriched, shared | Integral component of membrane, plasma membrane |
| PF3D7_1449000 | Gamete egress and sporozoite traversal protein, putative (GEST) | 0 | Y | Gametocyte-specific | Endoplasmic reticulum, microneme, osmiophilic body, implicated in gamete egress |
| PF3D7_1014300 | Conserved protein, unknown function | 8 | | Gametocyte enriched, shared | Cytoplasm, membrane |
| PF3D7_1348000 | Conserved Plasmodium protein, unknown function | 2 | | Gametocyte-specific | Cytoplasm, microtubule |
| PF3D7_1324600 | Conserved Plasmodium protein, unknown function | 3 | | Gametocyte-specific | Cytoplasm, membrane |

*TM*: Transmembrane domains (TMHMM: PlasmoDB, PMID1115261. Credits: Anders Krogh, Bjorn Larsson, Gunnar von Heijne, and Erik L.L. Sonnhammer), *SP*: Signal peptides (SignalP: PlasmoDB. PMID15223320. Credits Bendtsen JD, Nielsen H, von Heijne G, Brunak S), *Parasite stage*: Gametocyte score category based on an analysis of previous gametocyte and asexual proteomic databases, as indicated in methods[53]. Details in Supplementary Data 1 and 5, *Predicted/known cellular component (homology)*: Gene ontological terms (curated or computed) from PlasmoDB version 28, or annotation for previously characterised proteins. For proteins without annotation or predicted function/location, domain prediction based on protein homology is presented in square brackets where available (HHPred protein prdomain prediction, The MPI bioinformatics Toolkit as an integrative platform for advanced protein sequence and structure analysis[78])

PfGEST is not retained on the gamete surface by 20 min after erythrocyte egress in *P. falciparum* (Supplementary Fig. 5B). Though two antibodies produced modest TRA in singlicate SMFA experiments at 10 μg/mL (generalised linear mixed model (GLMM): maximum 49.3% TRA, 95% CI: 31.0–62.8, $p < 0.001$), this was not replicable and showed no improvement with concentration (Supplementary Fig. 4). These data indicate that though naturally acquired PfGEST Ab appear to be biomarkers of TRA, there is no current evidence to support a mechanistic role in TRA.

The 13 potentially surface expressed proteins were represented by 39 targets on the microarray, of which 16 were statistically significantly associated with TRA in the SMFA. Though antibody responses to most gametocyte proteins decrease with age, responses to these 16 targets shared a pattern more similar to Pfs48/45 and Pfs230; breadth of response to these targets increased with age ($p = 0.007$), whereas magnitude of response, though highest in older individuals, showed no age-dependent association (Supplementary Fig. 6). Gametocyte-positive individuals recognising ≥4/16 of these targets (4 being the 75th percentile of breadth of response to these targets) showed lower infectiousness (LogR: OR 0.36 (95% CI: 0.23–0.56), $p < 0.001$) and infection rates (OR 0.32 (95% CI: 0.16–0.62, $p = 0.001$) in the DMFA (Fig. 3a, b). These patterns remained significant when Pfs48/45 or Pfs230 seropositive individuals were excluded (infectiousness: LogR, OR 0.34 (95% CI: 0.20–0.58), $p < 0.001$; infection rate: OR 0.31 (95% CI: 0.15–0.64), $p = 0.001$).

## Discussion
In this study, we assessed antibody-mediated transmission-reducing activity (TRA) in individuals naturally exposed to *P. falciparum* and identified associated antibody responses. A

minority of samples (3.3%) demonstrated strong and reproducible levels of TRA in controlled in vitro assessments with cultured *P. falciparum* gametocytes; high-level TRA was associated variably with antibody responses to gametocyte antigens Pfs48/45, Pfs230 and 43 newly described proteins. These antibody responses were associated with reduced mosquito infection rates in feeding assays with local mosquito species on infected, gametocytaemic individuals. Although not all of the identified antibody responses will be causally involved in TRA, we provide the first direct evidence for independent roles of antibodies specific to Pfs48/45 and Pfs230 in naturally acquired TRA. The TRA of mAb specific to one of the novel TRA biomarkers, PfGEST, indicate that PfGEST antibodies elicited by *P. falciparum* exposure are likely to be non-functional biomarkers of TRA.

The transmission of *P. falciparum* to mosquitoes is influenced by numerous factors, including gametocyte density[60], sex ratio[61] and human host factors that are currently poorly characterised. A direct effect of naturally acquired antibodies on transmission efficiency is suggested by the observation that mosquito infection rates generally increase in mosquito-feeding experiments on naturally infected gametocyte carriers when autologous plasma is replaced by control serum[62]. Antibody responses to Pfs48/45 and Pfs230, established target antigens for transmission-blocking vaccines[15,36], are naturally acquired upon gametocyte exposure and have previously been statistically associated with TRA as measured in the SMFA[10,21–28]. The strength of these associations is variable and sometimes absent[29]; TRA is often observed in individuals without measurable α-Pfs48/45 or α-Pfs230 antibodies[10,23,25,28,30,31]. We performed SMFA experiments on purified IgG samples from 648 malaria-exposed donors and confirm previous associations between TRA and α-Pfs48/45 (OR 4.4, 95% CI: 1.5–12.9) and α-Pfs230 antibodies (OR 7.9, 95% CI:

2.8–22.8). Using a unique sample set with transmission data from DMFA from three endemic settings, we further provide the first evidence that α-Pfs48/45 and α-Pfs230 antibodies are associated with reduced transmission efficiency during natural infections. Both the likelihood of infecting any mosquitoes and the proportion of infected mosquitoes were significantly lower for individuals with α-Pfs48/45 and α-Pfs230 antibodies. For the first time, we formally demonstrate that naturally acquired antibodies to these gametocyte antigens can independently inhibit mosquito infection after affinity purification.

Despite the important role of antibodies to these two prefertilisation antigens, 45% of plasma donors with high-level TRA had no measurable α-Pfs48/45 or α-Pfs230 antibodies, and we observed significant levels of TRA after IgG from a selection of donors was depleted of α-Pfs48/45 and α-Pfs230 antibodies. Our affinity purification experiments were based on the binding of antibodies to epitopes present in the Pfs48/45-10C and 230CMB vaccine constructs, as the proteins' complex conformational structures so far preclude their production as full-length proteins. We thus captured the most potent transmission-reducing antibodies (Pfs48/45 epitopes 1–3[21,22,36], Pfs230 epitopes within the C region[15]) but not necessarily all antibodies to these proteins. Monoclonal antibodies against epitope 5, which would not bind to the Pfs48/45-10C protein, have been associated with weak TR activity[63]. However, given their lower efficacy and the immunodominance of epitopes 1–3 in naturally immune sera[21] it is highly unlikely that α-Pfs48/45 antibodies to epitopes other than epitopes 1–3 would have contributed considerably to TRA. This is also supported by the finding that antibodies to epitope 5 are only observed in the presence of antibodies to epitopes 1–3[21]. Because all known α-Pfs230 antibodies are complement dependent[64,65], whereas TRA in the antibody-depleted fraction is complement independent, a role for residual α-Pfs230 antibodies (responsive to epitopes outside the aa. 444–730 region of Pfs230 represented by the 230CMB product) is similarly unlikely. Crucially, IgG depleted of antibodies to Pfs48/45 epitopes 1–3 recognised surface antigens of a knockout parasite line that fails to express Pfs48/45 and Pfs230, formally demonstrating recognition of other target antigens by immune sera. Our findings of TRA in the absence of measurable α-Pfs48/45 or α-Pfs230 antibodies, together with persisting recognition of gametes that do not express these two proteins, and functional TRA after depleting IgG of antibodies to these two targets, thus strongly suggest that responses to other antigens may contribute to naturally acquired TRA.

Using a protein microarray, we aimed to identify whether responses to other gametocyte targets were associated with TRA. This array was enriched for putative gametocyte surface antigens but surface expression or gametocyte specificity were not prerequisites for inclusion on the array. Our aim was to determine an immune signature of TRA that could improve our understanding of transmission efficiency from natural infections. We therefore included markers of gametocyte exposure as well as general markers of recent malaria infection. We observed that responses to 43 proteins were statistically significantly associated with high-level antibody-mediated TRA. Responses to these antigens were strongly predictive of transmission efficiency from natural *P. falciparum* infections with diverse gametocyte strains and locally relevant mosquito species. Whilst further studies are needed to assess the generalisability of this signature of TRA across endemic settings, the strong association of our SMFA-based signature of TRA with reduced transmission potential during natural infections is very promising. Future studies should aim to assess the kinetics of antibody responses to these antigens and their association with TRA using longitudinal cohorts, with well-defined clinical, parasitological and infectivity data.

The microarray proteins were produced in an *Escherichia coli* based in vitro transcription–translation (IVTT) system. As conformation is not verified for all proteins, those requiring post-translational modifications will likely not have been expressed in their native conformation. This was exemplified by Pfs48/45 that has a known complex structure[36,66] and showed minimal reactivity or discriminative power when printed on the array as an IVTT product (Supplementary Data 3). This apparent disadvantage of the IVTT system will plausibly have affected our results for other conformational proteins, so the list of proteins in Table 3 (and Supplementary Data 5) must be treated as a 'rule in' rather than 'rule out' selection for proteins associated with antibody-mediated TRA.

Among the proteins we identified as having TRA-associated antibody responses, 13 novel proteins have sequence characteristics suggestive of surface expression and may be targets of TRA. Previous data indicate that two of the proteins found to be biomarkers of TRA in our analysis, PfGEST and Pf11-1, were previously shown to have important roles in gamete egress from the erythrocyte in mosquitoes[58,59]. Interestingly, Pf11-1 mAbs are able to reduce transmission, possibly by disabling full gamete egress[58]. Our SIFA data with humanised mAbs indicate that PfGEST has no clear surface expression in *P. falciparum* 20 min after activation, which may explain their lack of replicable TRA in mosquito-feeding assays compared to similar concentrations of α-Pfs230-specific mAb. As there is only a narrow window of opportunity for α-Pf11-1 and α-PfGEST antibody binding during gamete egress, activity against gamete surface proteins with roles in fertilization is a more biologically plausible mode of antibody-mediated TRA. Future experiments may identify which of the 13 other TRA-associated plausible antibody targets in Table 3 are expressed at the gamete surface. Our findings identify the further characterisation of functional and non-functional gametocyte antibody responses in individuals with naturally acquired TRA as a research priority. Although the prevalence of high-level TRA decreases with age, an opposite trend is observed for the magnitude and prevalence of antibodies to Pfs48/45 and Pfs230 and to newly identified gametocyte antigens, in spite of their overall association with TRA (Supplementary Fig. 6). A better characterisation of these antibody responses in relation to functional activity is needed and may include a detailed assessment of antibody isotype, affinity and complement-binding activities.

Regardless of their functionality, our data indicate that possession of antibody responses to a group of *P. falciparum* proteins increases the likelihood of antibody-mediated transmission inhibition. Combined with assessment of α-Pfs48/45 and α-Pfs230, this signature of TR immunity may improve our understanding of natural malaria transmission dynamics.

## Methods

**Study participants**. Serum was collected from individuals recruited to epidemiological studies in Burkina Faso, Cameroon and the Gambia (Table 1), as well as from Dutch migrants with a history of residence in *P. falciparum* malaria-endemic regions, who reported having been diagnosed with clinical malaria infection. Details of sample collection and recruitment criteria have been published for all samples from individuals living in endemic areas, but are provided here as a summary. Parasitological status was defined by microscopy acknowledging that this lacks the sensitivity to detect many low-density infections; i.e., microscopy negative samples were not treated as an unexposed control.

*Cameroon*. Samples ($n = 140$) were collected between 2011 and 2015 from individuals aged 5–16 years, during surveys at primary schools in the Mfou district, 30 km from Yaoundé. Data from these individuals have in part been presented previously[42,43], but the remainder were provided by Isabelle Morlais (IRD, Montpelier) without prior publication. Participants were primarily *P. falciparum* gametocyte carriers identified among asymptomatic children by microscopical screening. Gametocyte density (median (IQR) = 30 (16/80)/µL, among gametocyte positives) was calculated against 1000 leucocytes, assuming a density of 8000 leucocytes per µL.

*Gambia*. Samples (*n* = 226) were collected from individuals resident in Farafenni and nearby villages, approximately 200 km east of Banjul, in 1992–1994[24,44], 1998 and 1999[45], 2001[46,47] and 2002[48]. Individuals were aged 1–17 years at the point of sampling. Participants were enroled if they were found to be asymptomatically infected with malaria parasites, based either on the observation of gametocytes[24,44], or asexual parasites[45–48] by microscopy. Gametocyte density (median (IQR) = 63 (10/390)/μL, among gametocyte positives) was either calculated by examination of 100 high powered fields, assuming that 1 gametocyte per field equated to 500/μL (threshold of 5 gametocytes/μL) as in Greenwood & Armstrong 1991[67], or by calculation against 1000 leucocytes, assuming a density of 8000 leucocytes per μL.

*Burkina Faso*. Samples were collected from residents in the villages of Laye and Dapélogo (*n* = 192), ~30 km northeast and north of Ouagadougo, in the Central Sudan savannah area[37,40,41], and from residents of Soumousso and Dande (*n* = 33), villages close to Bobo Dioulasso[38,39]. The age of study participants ranged from 1.8 to 74 years. Recruitment was either random[37,41], or based on the observation of gametocytes[38–40] or asexual parasites[40] by microscopy. Gametocyte density (median (IQR) = 64 (32/128)/μL, among gametocyte positives) was calculated against either 500 or 1000 leucocytes, assuming a density of 8000 leucocytes per μL.

*Netherlands*. Dutch individuals were recruited by clinical staff at Radboud university medical centre in Nijmegen, the Netherlands, and had either recently returned ( < 6 months) from extended periods in endemic areas (*n* = 8) or had returned more than a year before sampling (*n* = 49). Age ranged from 26–84 years. Those who had returned more than a year before sampling had lived in areas with endemic *P. falciparum* transmission for longer than 15 years and reported between 2 and >80 clinical malaria episodes during that time. For these individuals malaria infection history was only available through verbal testimony, whereas 3 out of 8 of the recently returned migrants had confirmed recent or current malaria infection. Samples were all collected between 1994 and 2014. As detailed previously, sera from some of these expatriates show strong and consistent TRA in the SMFA[68].

All samples were collected after written informed consent was obtained from participants and/or their parent(s) or guardian(s). Ethical clearance was provided by the National Ethics Committee of Cameroon, the Gambian Government/ Medical Research Council Joint Ethics Committee, the Ethical Review Committee of the Ministry of Health of Burkina Faso and the Centre MURAZ ethical review committees, the London School of Hygiene & Tropical Medicine ethics committee and by the Radboud University Medical Center ethical committee.

**Sample preparation**. For SMFA, IgG was purified from all plasma samples using Protein G HP Spintrap (GE Healthcare, GE Uppsala, Sweden), and concentrated to the original serum volume using Vivaspin 20 centrifugal concentration columns (Sartorius AG, Goettingen, Germany) following the manufacturer's instructions. Antibody purification was only attempted for samples with sufficient serum volume for duplicate SMFAs (180 μL). For the serological assays, an additional 30 μL of serum was retained if sample volume permitted. Because of the high serum volumes necessary for the antibody depletion/purification experiments, these were performed on only six samples with transmission-blocking antibodies. Three of these six were processed by protein microarray. IgG for depletion/purification was purified from 1 mL of serum using protein G high performance affinity columns (HiTrap™ Protein G HP 1 mL, GE Healthcare, Uppsala, Sweden), and then adjusted to the original serum volume for testing in the specific Pfs48/45 and Pfs230 affinity columns using Vivaspin 20 centrifugal concentration columns (Sartorius AG, Goettingen, Germany). Overall, 300 μL was immediately removed for SMFA and serological assays, leaving 700 μL of total IgG at physiological concentration for affinity column purification. Purification was repeated as required to allow for antibody eluate concentration.

**Direct membrane-feeding assay (DMFA)**. DMFA was performed during the original studies for 498/579 individuals from endemic areas, as described in the separate study protocols using identical glass mini-feeders[54] (Coelen Glastechniek, Weldaad, the Netherlands) and the same experimental protocol[69]. Of the 498 individuals with DMFA data, 494 had microscopy-based estimates of asexual and gametocyte density; only gametocyte positives (*n* = 366) were used in our DMFA analysis. All sites used colonies sourced from eggs caught in local water sources. Colonies were of *A. gambiae s.s.* (>90%) (The Gambia[24,44–48]), *A. coluzzii* (Cameroon[42,43] and Bobo Dioulasso in Burkina Faso[38,39]) or a comprised mixture of *A. coluzzii*, *A. gambiae s.s.* and hybrid forms (other Burkina Faso sites[37,40,41]). Locally adapted mosquito colonies from all sites are available upon request. Data presented in the current manuscript are the proportion of infectious individuals and percent of mosquitoes infected with oocysts after feeding on whole blood[69].

**Standard membrane-feeding assay (SMFA)**. Mosquitoes used for the assessment of antibody TRA in the SMFA were three to 5 day old female *Anopheles stephensi* (Sind-Kasur Nijmegen strain)[70], which were reared at 30 °C and 70–80% humidity while exposed to a 12 h day/night cycle. Mature (stage IV-V) *P. falciparum* gametocytes (either NF54[54], or transgenic NF54HT-GFP-Luc[71], depending on the preferred readout) were obtained from an automated tipper system

(0.3–0.5% gametocytes, 2% haematocrit) and prepared with packed red blood cells as previously described[9,54]. Both gametocyte strains can be obtained from the MR4 Malaria Research and Reference Reagent Resource Center (https://www. beiresources.org/MR4Home.aspx). For the preparation of mosquito blood meals containing test antibodies, 90 μL of serum IgG was added to 90 μL of freeze-dried foetal calf sera (FCS) and diluted into 35 μL human serum containing active complement. This antibody/serum mix was added to the gametocyte/red blood cell mix to a final volume of 270 μL, and kept at 37 °C until it was provided to mosquitoes in the membrane-feeding apparatus. Antibodies were therefore provided to mosquitoes at ~0.66* their physiological concentration (assuming a 1:1 plasma/cell ratio). Each SMFA run was performed to test up to 24 antibody samples simultaneously, including two control mosquito groups (against which the TRA of test samples was calculated). For control samples 90 μL of freeze-dried FCS was dissolved in 90 μL milliQ and added to the same gametocyte/red blood cell mix (270 μL total volume) as was used for the other feeds the run.

Prior to blood feeding, male mosquitoes were separated from females by aspiration. Unfed and partially fed female mosquitoes were removed from their resting containers after blood feeding. Fully engorged mosquitoes were maintained at 26 °C and 70–80% humidity. Mosquito infection status was determined differently for the NF54 and NF54HT-GFP-luc parasite strains. For experiments with the native NF54 strain, an average of 19.5 mosquitoes (range: 7–22) were dissected at day 7 post infection (PI). Midguts were stained in 1% mercurochrome and oocysts were counted by expert microscopists. For experiments with the transgenic NF54HT-GFP-luc strain, mosquitoes were killed on day 8 PI by freezing, and 20 were homogenised for each experiment in two pools of 10. Mosquito homogenates were well mixed and three samples per pool were assayed for bioluminescence by lysis, and addition of luciferase assay substrate using a Luciferase assay kit (Luciferase assay system (E1501), Promega, USA), according to the manufacturer's instructions and as previously described[9].

For experiments with the native NF54 strain, TRA was analysed as the percent difference in oocyst number between test and control mosquitoes. For experiments with the transgenic NF54HT-GFP-luc strain, TRA was calculated as the average percent difference in luminescence intensity for two pools of 10 mosquitoes between test and controls mosquitoes, as described[9]. TRA <0% was observed, but was generally within the normal range due to natural variation in oocyst intensity between mosquito groups. For individuals with TRA <0%, median TRA was -41.8% (IQR: −80.9–16.5). For experiments with replication (e.g. the affinity purification experiments) 95% CI around the TRA estimate were calculated from luminescence or microscopical readouts for multiple feeds using generalised linear mixed models (GLMMs)[72]. Choice of readout (and therefore of gametocyte strain) varied over the time that the SMFA's were being performed, and makes no difference to the integrity or interpretation of TRA, only to its efficiency[9]. Overall, ≥90% reduction in oocyst/luminescence intensity in test mosquitoes relative to controls was classified as evidence of high-level TRA, which previous data indicate is highly replicable[68]. Samples from all individuals with ≥80% TRA were tested in duplicate to ensure robustness of the functional phenotype and the mean value of duplicate experiments was used to classify individuals as having high-level TRA (≥90% average reduction) or no evidence of TRA (<10% average reduction).

**Recombinant protein production for serological assays**. Recombinant, correctly folded proteins were used for antibody depletion/purification experiments and for enzyme-linked immunosorbent assays (ELISA). For all assays assessing Pfs230-specific antibodies, we used the transmission-blocking vaccine candidate 230CMB (developed and purified by Fraunhofer USA Center for Molecular Biotechnology), which corresponds to amino acids 444–730 of the Pfs230 protein, and comprises the protein's pro-domain and the first of fourteen cysteine motif domains[15]. For assays with Pfs48/45, two protein constructs were used. For antibody depletion/ purification, we used the chimeric R0.10C protein[36], which comprises 10 cysteines spanning epitopes 1–3 of the Pfs48/45 C-terminal domain, fused to the GLURP R0 domain[36,40]. Antibodies specific for GLURP R0 have no effect on *Plasmodium* transmission to mosquitoes (Supplementary Table 2)[9]. For non-functional serological assays (ELISA, Microarray), the 10C protein was cleaved from the R0.10C as described previously[40].

**Monoclonal antibodies**. SMFA data using mAb specific to GLURP R0, Pfs48/45 (85RF45.1), Pfs230 (2A2) and PfGEST are presented in Supplementary Fig. 4 and Supplementary Table 2. Full details for GLURP R0 and Pfs48/45 (85RF45.1) mAb production and SMFA is provided in Stone et al.[9] Details of Pfs230 mAb production and SMFA are presented in Roeffen et al. 1995[65]. PfGEST mAbs were produced by the Hypothesis-driven Pre-erythrocytic Antigen Target Identification Consortium (HPATIC). Briefly, HKL v2.0 Kymice™ containing human variable region genes (Kymab) were immunised with recombinant protein expressed in the wheat germ cell-free system (CellFree Sciences) adjuvanted with Montanide ISA 720. Non-antigen selected splenocytes were sorted into 96 well plates, one cell/well, to generate single-cell mRNA material (Atreca). Atreca's Immune Repertoire Capture (IRC™) provides Single-Cell mRNA Barcode-Sequencing with high throughput and accuracy. This platform captured full-length sequences, enabling detection of somatic mutations across the entire antibody variable region. Natively paired heavy and light chain IgG variable regions were analysed, and constant region isotype assignments were made which were then produced via gene

synthesis and recombinant expression as fully human antibodies (LakePharma). Three mAbs were tested in duplicate SMFA titration experiments. SMFA was conducted according to previously published protocols[9]. In all mAb-based SMFA, the endpoint was oocyst quantification by manual dissection.

**Pfs48/45 and Pfs230 antibody purification.** To purify antibodies specific to the Pfs48/45 and Pfs230 proteins from transmission-reducing sera, affinity columns were created by coupling 1 mg of R0.10C or 230CMB to N-hydroxysuccinimide (NHS)-activated high performance affinity chromatography columns (HiTrap™ NHS-activated HP 1 mL, GE Healthcare, Uppsala, Sweden); coupling efficiency was >80%.

For the extraction of antibodies from human serum, antibodies were extracted first using the R0.10C column, and then using the 230CMB column, as follows. The R0.10C column was equilibrated with 2 mL of binding buffer (PBS (1×) before 700 μL of total IgG (at physiological concentration, fraction 1 in Table 2) from transmission-blocking sera was added. The flow-through was collected and the column washed with 12 mL of PBS. Overall, 5 mL of elution buffer (0.15 M NaCl, 0.1 M glycine/HCL, pH 2.5) was applied to the column for the collection for R0.10C-specific antibodies. After elution, the purification procedure was repeated using the flow-through from the first purification, and the antibody eluates from both runs were pooled. The final flow-through and combined antibody eluates (fraction 2a in Table 2) were concentrated to 700 μL using Vivaspin 20 centrifugal concentration columns (Sartorius AG, Goettingen, Germany). Overall, 300 μL of the flow-through was removed for SMFA (fraction 3 in Table 2) and serological assays, and the remaining 400 μL was processed over the equilibrated 230CMB column as for the R0.10C column, except that the final flow-through's (fraction 5 in Table 2) and eluates (fraction 4a in Table 2) were concentrated to 400 μL. Depletion of specific antibody in the flow-through's of each column was confirmed with ELISA. Antibodies (from eluates or flow-troughs) were processed in the SMFA in two biological replicates, as for total IgG with >80% TRA in the larger SMFA screen. For three samples, larger starting volumes were used to allow antibody eluates to be concentrated nine times, to examine the effect of specific concentrated antibody on transmission-reducing activity (fractions 2b and 4b in Table 2).

The possibility that TRA from the column fractions was due to cytokines or reactive ROx/RNOx species, which in circumstances of inflammatory crisis can inhibit parasite development in mosquitoes[73], was excluded both by our use of protein G binding columns for IgG purification for all samples, and our use of 30 kDa filters for antibody concentration prior to mosquito-feeding.

**Surface immuno-fluorescence assays (SIFA).** The Pfs48/45 KO NF54 *P. falciparum* line was produced and characterised by van Dijk et al[3]. and subsequently assessed by Eksi et al.[12] Eksi et al. showed that Pfs230 is produced but not retained on surface of Pfs48/45 disruptant gametes. Wild-type (WT) and Pfs48/45 KO gametocytes were generated using standard protocols, as described above. Culture media Pfs48/45 KO line was supplemented with 2 μM pyrimethamine. Briefly, gametocytes were allowed to activate at room temperature for 20 min (for mAb SIFA) or 1 h (for human IgG SIFA) in the presence of foetal calf serum (FCS), and gametes were prepared for microscopy by washing with PBS supplemented with 0.5% FCS and 0.05% sodium azide. Antibodies were added to gamete preparations in the same PBS/FCS buffer; mAbs at 5 μg/mL; human IgG/sera at dilutions of 1:20. Secondary antibodies were Alexa Fluor TM 488 goat anti-human IgG (H+L), Alexa Fluor TM 488 chicken anti-mouse IgG (H+L), and Alexa Fluor TM 488 goat anti-rat IgG (H+L); all were added at dilutions of 1:200. All gamete preparations were incubated with secondary antibody at room temperature for 1 h.

**ELISA.** Antibodies specific to the Pfs48/45-10 C and Pfs230 (230CMB) proteins were quantified in the ELISA exactly as described previously[40]. Optical density (OD) values were normalised between assay plates by adjustment relative to a consistent point in the linear portion of a standard curve of serially diluted highly reactive human sera. Cut-offs for positivity were determined from normalised OD values using maximum likelihood methods to define Gaussian populations of low and high responders as described previously[74]. For Dutch migrants who had returned from endemic areas ≥1 year before sampling, a seropositivity threshold was calculated as the mean+3 standard deviations of the averaged normalised OD values of eight naïve European control sera.

**Protein microarray.** The selection of proteins to be cloned and printed on the arrays was made primarily based on observed expression by stage V *P. falciparum* gametocytes in a single recent proteomic analysis[52]. To ensure that the array included most potential antibody targets, we preferentially included proteins that had transmembrane domains, signal peptides or GPI anchors. To ensure that the majority of proteins that were highly abundant or 'specific' to mature gametocytes were included on the array, we used mass spectrometry data from the analysis of the stage V gametocyte proteome[52], alongside proteomic data from purified trophozoites[1] and schizonts (Supplementary Data 6, PRIDE accession number: PXD008250) generated in the same laboratory. Gametocyte and asexual stage-specific expression data sets were generated in relative expression values to

compare protein abundances (label-free quantitative (LFQ) values) between the stages, which are presented in Supplementary Data 1 as a fold-change value.

We chose to also include proteins that were potential markers of gametogenesis identified by Silvestrini et al.[1] These proteins we identified as being expressed primarily by early gametocytes, or shared between stages (proteins for 32/59 were present in the stage V database), but some were more abundant in mature gametocytes or other parasite life stages (Supplementary Data 1). In addition to these markers, we included four putative markers of asexual parasite exposure: PF3D7_1036000 (Merozoite surface protein 11 (MSP11)), PF3D7_0711700 (erythrocyte membrane protein 1, PfEMP1 (VAR)), PF3D7_0731600 (acyl-CoA synthetase (ACS5)), and PF3D7_0423700 (early-transcribed membrane protein 4 (ETRAMP4)), based on work by Helb et al.[51]

Details of the 315 proteins included on the array are given in Supplementary Data 1. Proteins with reference sequences longer than 1000 amino acids were split into multiple fragments (overlaps of at least 17 a.a) for cloning, in vitro protein expression and printing. Because the basic criteria for the selection of proteins for microarray construction was based on data generated from a single laboratory, we cross-referenced our protein list against a cross-study analysis of gametocyte specificity[53]. In short, the specificity of any protein for the gametocyte stage was scored by determining how often it had been detected across 11 proteomic analyses (listed in Supplementary Data 1: 3 of gametocytes only, 3 of asexual stages only, and 5 of both asexuals and gametocytes). Proteins were binned from low to high abundance and weighted according to the retrieval rates of proteins in two curated lists of 'gold standard' gametocyte and asexual genes, consisting of genes that are known to be specific for either asexual stages (*n* = 45) or gametocytes (*n* = 41). High expression of gametocyte gold standard proteins with concurrent absence of non-gametocyte gold standard proteins resulted in a high gametocyte score, calculated from the fraction of retrieved gametocyte genes over retrieved non-gametocyte genes. All scores were log-transformed and summed over all data sets. Gametocyte scores were categorised using the scores' distribution of the gametocyte gold standard as follows: scores above the first quartile (9.69) were considered gametocyte-specific. Proteins scoring at least as high as the lowest scoring gametocyte gold standard representative (-2.46) were considered as enriched in gametocytes. Proteins with lower scores are not specific to a certain life stage and have some evidence for expression in gametocytes. All proteins scoring lower than the median of the asexual gold standard (-18.98) were considered asexual specific. This analysis confirmed that 228 of the 315 proteins had consensus evidence for enrichment in gametocytes (scores >−2.46), whereas 284/315 had conservative evidence for expression in gametocytes (peptides present in at least one gametocyte proteomic database, score >−10). 312/315 were present in at least one gametocyte proteomic database, giving rise to a score >−18.93. Scores for all proteins on the array are included in Supplementary Data 1. To avoid further curation and potential bias, no proteins were excluded prior to analysis.

Proteins were expressed using an in vitro transcription and translation (IVTT) system, the *Escherichia coli* cell-free Rapid Translation System (RTS) kit (5 Prime, Gaithersburg, MD, USA). A library of partial or complete CDSs cloned into a T7 expression vector pXI has been established at Antigen Discovery, Inc. (ADI, Irvine, CA, USA). This library has been created through an in vivo recombination cloning process with PCR-amplified CDSs, and a complementary linearised expressed vector transformed into chemically competent *E. coli* was amplified by PCR and cloned into pXI vector using a high-throughput PCR recombination cloning method described elsewhere[50]. Each expressed protein includes a 5′ polyhistidine (HIS) epitope and 3′ hemagglutinin (HA) epitope. After expressing the proteins according to manufacturer instructions, translated proteins were printed onto nitrocellulose-coated glass AVID slides (Grace Bio-Labs, Inc., Bend, OR, USA) using an Omni Grid Accent robotic microarray printer (Digilabs, Inc., Marlborough, MA, USA). Each slide contained eight nitrocellulose 'pads' on which the full array was printed, allowing eight samples to be probed per slide. Microarray chip printing and protein expression were quality checked by probing random slides with anti-HIS and anti-HA monoclonal antibodies with fluorescent labelling. Each chip contained 28 IVTT negative control targets (for data normalisation), and 48 IgG positive control targets (for quality control).

For analysis of antibody reactivity on the protein microarray, serum samples were diluted 1:200 in a 3 mg mL$^{-1}$ *E. coli* lysate solution in protein arraying buffer (Maine Manufacturing, Sanford, ME, USA) and incubated at room temperature for 30 min. Chips were rehydrated in blocking buffer for 30 min. Blocking buffer was removed, and chips were probed with pre-incubated serum samples using sealed, fitted slide chambers to ensure no cross-contamination of samples between pads. Chips were incubated overnight at 4°C with agitation. Chips were washed five times with TBS-0.05% Tween 20, followed by incubation with biotin-conjugated goat anti-human IgG (Jackson ImmunoResearch, West Grove, PA, USA) diluted 1:200 in blocking buffer at room temperature. Chips were washed three times with TBS-0.05% Tween 20, followed by incubation with streptavidin-conjugated SureLight P-3 (Columbia Biosciences, Frederick, MD, USA) at room temperature protected from light. Chips were washed three times with TBS-0.05% Tween 20, three times with TBS, and once with water. Chips were air dried by centrifugation at 1000×*g* for 4 min and scanned on a GenePix 4300A High-Resolution Microarray Scanner (Molecular Devices, Sunnyvale, CA, USA). Target and background intensities were measured using an annotated grid file (.GAL).

Raw microarray protein target and local background fluorescence intensities, target annotations and sample phenotypes were imported and merged in R

(Foundation for Statistical Computing, Vienna, Austria), where all subsequent procedures were performed. Foreground target intensities were corrected for local background using the backgroundCorrect function of the *limma* package[75]. Next, all corrected values were transformed using the base 2 logarithm. The data set was normalised to remove systematic effects by subtracting the median signal intensity of the IVTT controls for each sample. As the IVTT control targets carry the chip, sample and batch-level systematic effects, but also antibody background reactivity to the IVTT system, this procedure normalises the data and provides a relative measure of the specific antibody binding to the non-specific antibody binding to the IVTT controls (a.k.a. background). With the normalised data, a value of 0.0 means that the intensity is no different than the background, and a value of 1.0 indicates a doubling with respect to background.

**Data analysis.** Quality control plots were made after each treatment of the microarray data, which included boxplots and density plots of probe intensity by study sample and probe type. Recombinant Pfs230 CMB was printed on the array for comparison to ELISA with the same protein, yielding a correlation co-efficient (Spearman's rank) of 0.44 ($p < 0.001$). A universal seropositivity threshold for reactivity to array proteins was established as the IVTT background plus 3 standard deviations, which equated to a log$_2$-transformed signal intensity threshold of 0.92 (reactivity was therefore defined as approximately twice that of the background). Because high-level transmission-reducing activity was uncommon in our sample set (22/649 samples had TRA ≥90%), antigens were defined as 'reactive' if they elicited seropositive responses in at least 1% of the study population (≥7 subjects), and unreactive antigens were included in all analyses. The universal cut-off was used only for defining proteins as reactive or unreactive, and for depicting antibody breadth for all microarray proteins.

Statistical analysis was conducted using R (Foundation for Statistical Computing, Vienna, Austria) or STATA 12 (StataCorp., TX, USA). The magnitude of antibody reactivity to each antigen on the microarray by individuals in different groups was tested by empirical Bayes-moderated *t*-tests[76], and by logistic regression after data binarisation using protein-specific cut-offs; cut-offs were generated using maximum likelihoods methods (which determines the junction between two Gaussian signal distributions for each protein that proteins cut-off point for positivity)[74]. For the latter, proteins were dis-included from the analyses if the mixture models failed to stably converge, or had to restart more than once. In both the Bayes and logistic models, *p*-values were adjusted to control the false discovery rate below 5% using the Benjamini–Hochberg (BH) method[57], and a finding was considered significant for FDR controlled *p*-values (q-values) <0.05. For analysis of TRA on a continuous scale, TRA was used as the log-transformed relative intensity, to normalise the data and avoid compression at 0%. For clarity, figures presenting TRA on a continuous scale present the linear TRA with a base of 0%. Linear models with adjustment for gametocyte density (determined by microscopy, and given as gametocytes/μL) were used to test for differences in continuous variables between groups. Logistic models adjusted for gametocyte density were used to test for the association of binary variables between groups, presenting odd-ratios and 95% CI. Differences in the magnitude of response to all proteins by groups of individuals (in contrast to the Bayes analysis, which compares responses to individual proteins by different groups of individuals) were assessed with t-tests or ANOVA. Differences in mosquito infection prevalence in the DMFA were analysed with logistic models adjusted for gametocyte density, while differences in mosquito oocyst prevalence were assessed with logistic models, adjusted for gametocyte density and with host (individual the mosquitoes were feeding upon) as a random effect. All boxplots are standard Tukey's whisker and boxplots, with Tukey's method for outlier determination. All DMFA-based analysis was performed on gametocyte-positive individuals only. Protein-domain predictions for proteins with TRA-associated responses were made using Inter-pro[77] or HhPred protein-domain prediction[78].

**Data availability.** The novel *P. falciparum* Schizont mass spectrometry proteomics data have been deposited to the ProteomeXchange Consortium (http://proteomecentral.proteomexchange.org) via the PRIDE partner repository[79] with the data set identifier PXD008250. The authors declare that all other data supporting the findings of this study are available within the article and its Supplementary Information, have been deposited in the DRYAD data depository (https://doi.org/10.5061/dryad.8bp05), or are available from the authors upon request.

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

## Acknowledgements

This work was supported by a Marie Curie Career Integration Grant from the European Community's Seventh Framework Programme (SIGNAL, PCIG12-GA-2012-333936), and by a VIDI fellowship from the Netherlands Organization for Scientific Research (NWO; Project Number 016.158.306). Data collection in Burkina Faso was supported by the Bill and Melinda Gates Foundation (AFIRM OPP1034789). PfGEST mAb were produced by the Hypothesis-driven Pre-erythrocytic Antigen Target Identification Consortium (HPATIC) and provided by PATH's malaria vaccine initiative. We thank the participants of the various trials contributing samples and data to this study, and the staff involved in these trials. We thank Jacqueline Kuhnen, Laura Pelser-Posthumus, Astrid Pouwelsen, Jolanda Klaassen, Maarten Eldering and Tonke Raaijmakers for their

role in sample preparation, mosquito breeding and conduct of the SMFA. We thank Sanna Rijpma for her assistance with the preparation of samples for proteomic analyses. We thank Kassie Dantzler and Siyuan Ma for their assistance with the microarray analysis. We thank Guido Bastiens for arranging sampling of Dutch migrants, and the missionaries for their agreement to participate in the current study by donating blood for serological and functional analyses. We also thank Jessica Chichester (Fraunhofer) for providing us with Pfs230 CMB.

## Author contributions

T.B., P.F., M.J., R.S. and W.S. conceived and designed the study. I.M., D.D., A.C., S.N., C. D., C.J.S. and A.L.O. provided plasma samples, and related malariometric and direct membrane-feeding data. M.v.d.V.B., G.-J.v.G, W.G, R.S.-S., K.L. and W.S prepared samples for, conducted and evaluated the standard membrane-feeding assays. T.B, M.M., S.K., P.F. and W.S. designed the microarrays. E.L., G.G. and E.S. provided proteomic data for selection of proteins for microarrays. J.P. and A.T. produced proteins for and printed microarrays. A.S. and W.S. performed the microarray assays. M.J., S.J. and W.S. performed serological analyses. W.R., S.S. and M.T. provided recombinant proteins. M.J. and W.S. performed the affinity purification experiments. M.J, R.M.d.J. and A.F.G. performed the SIFA experiments. J.C. and W.S. prepared the microarray data, and performed the microarray analyses. J.B. and W.S. performed all data analysis. L.-M.K. provided bioinformatics support, and gametocyte enrichment data. W.S. wrote the first draft of the manuscript, and performed all data analysis. C.D., T.B., M.J. and W.S. wrote the final draft of the manuscript. T.B. was the principal investigator on this project. All authors read and edited drafts of the manuscript and approved its final iteration.

## Additional information

**Competing interests:** P.F. owns stock and is a board member at Antigen Discovery, Inc. The remaining authors declare no competing financial interests.

