## [Peer Review File · Nature Communications]

Reviewers' comments:

Reviewer #1 (Remarks to the Author):

Stone, et al.-- "Unravelling the immune signature of Plasmodium falciparum transmission reducing immunity"

The authors describe a large body of data that "demonstrates that the efficiency of malaria transmission from humans to mosquitoes is modulated by host antibody responses to gametocyte antigens", not only to the P230 and P48/45 but also to a larger repertoire of novel parasite targets. Although at first read this is not a new concept, as the association has been described since the late 90's, the authors now provide the missing evidence supporting a causal association of anti-P230 or anti-P48/45. This is interesting and partially ironic since P230 and P48/45 have already moved forward on several occasions towards vaccine development. It was assumed that there was enough causal rationale to support vaccine development by now. What is really "new" is the discovery of humoral responses to additional Plasmodium proteins. However, not only does the clarity of the manuscript require improvement, but very simple validation experiments are missing to support the discovery. The methods and analytical approaches, as described, appear a bit flawed in design, but perhaps the rationale is simply missing? The manuscript describes some potentially exciting gems as a result of their work, but a compelling case is still needed. Overall, the manuscript details a comprehensive and almost thorough set of experiments, results and analyses. The addition of a few experiments and key information to "fill in the blanks" for a broader audience is required before further consideration for publication.

Major comments:

1. It is difficult to understand why antibodies that were reactive with a subset of the novel proteins were not tested for IFA staining of stage V gametocytes or gametes following induction of gametogenesis. The authors clearly have all the reagents in hand and it would go a long way to providing the data to show "relevance" of some of the presumed targets identified by the protein array. As it stands the list of novel candidates remains long and the rationale for their selection in the first place is missing. The authors should have prioritized the analysis of a subset of targets, especially those that are predicted NOT to be on the cell surface of gametocytes or gametes. It is odd that "presence of a transmembrane membrane domain" was not considered as a potential red-herring, since there are several proteins that are not secreted to the cell surface that have such a domain signature, e.g., ER and Golgi proteins, as obvious examples. Inner membrane proteins can also fall into this category. The authors included 19/42 proteins in Table 3 that are likely not to be the targets of antibody-mediated transmission-reducing activity as they are not on the cell surface. Mitochondrial, nuclear and apicoplast proteins have been included? Are the authors suggesting that a new blocking mechanism is in effect? Also, the inclusion of proteins in the list despite not meeting their selection criteria is puzzling. If there is a null prediction for any relevance to the proposed biological mechanism of antibody-mediated inhibition, then why include it in the list in a main table? It is as if the list was not checked again. For example, randomly selecting PF3D7_0505000 identifies a protein with no predicted location (according to Table 3) but the potential presence of 4 transmembrane domains

(supplemental Table 4) likely keep it in the list (though this is a soft inclusion-criterion). Yet, the fact that it has a Dos2-interacting transcription regulator of RNA-Pol-II domain (PlasmoDB) should have potentially precluded it from the main table. It is recognized that the authors would like to demonstrate associations between transmission-reducing activity and antibody recognition of a protein following the probing of a protein array, but some caution is merited here and a closer curation of the data is needed (with the accompanying plausible rationale). Including a table of data highlighting associations with potentially limited biological relevance should be avoided.

2. The authors should be commended on identifying the major drawbacks of the protein array (a major approach in their study). However, these statements are not enough to explain the results nor does it excuse potentially misleading conclusions (point 1). Some quality control of the results in the form of validation experiments is necessary. It is a bit of a disservice to the community if the data are simply put out there as if they are already leading vaccine candidates for consideration. The authors should recognize that they are an influential body to the larger malaria community and the data need to be carefully presented so that others do not waste time and valuable research funds chasing red herrings.

3. What level of stringency was applied in the probing of the protein arrays? No serum dilutions (dose-response curve) were performed. No orthogonal validation using a different assay was performed to confirm the read out? What was the density of proteins or peptides of a given protein per spot? How were the "multiple fragments" or domains of a given protein identified for representation on the array? What was the rationale behind the selection and how was the rationale demonstrated to be correct or reasonable? P48/45 was not included in Table 3. Is this not considered a positive control for the assay? The array described in the manuscript appears to be new and therefore the same level of measurement to ensure rigor and robustness needs to be applied.

4. It seems that not all of the DMFAs were performed in the same manner. It is assumed that the whole blood and serum-replacement experiments are all from the same donor. If this is not the case across all endemic sites that performed the DMFA, then this should be made clearer to the reader. Also, did all the sites performing DMFAs use the same operational cut-off for enrollment of an individual in the study in terms of gametocyte density or age? Did they all use the same mosquitoes? In clarifying these sections, the following question should be addressed: "would whole blood DMFAs change the prioritization of the 41 proteins?"

5. What does "All the following analyses are adjusted for gametocyte density at the moment of DMFA" mean? As written, these appear to be "convenience samples", not a study done specifically for the project described in this manuscript, with the exception of a few from Cameroon. What was the gametocyte density (operational definition) at the time of the DMFA in each site?

6. What was the baseline WBC count used for all microscopic evaluations? The published literature suggests that different sites use different standards for accuracy in counting parasites in blood smears. At what point is a slide considered gametocyte-negative? Is it counted against 200 WBC or 1000 WBC? How do such differences, if they are indeed performed differently, going to affect the overall analyses?

7. No human IgG subclass assessment was conducted?

8. Approximately 3.3% (22/648) samples had potent TRA. What age group? Which

countries? Similar across the board? Is TRA $\geq 80\%$ a meaningful benchmark? As written, it is difficult to follow what constitutes a meaningful biological phenomenon in terms of reduction of oocyst numbers in mosquitoes. The goal, as stated in the text is to find "biomarkers of broader sexual-stage immunity". Would the small proportion of high blockers in the entire dataset of 648 from different sites experiencing different malaria exposure detrimental to the overall analysis and subsequent conclusions?

9. The following is not very clear at initial read. "On a continuous scale, TR activity was not associated to gametocyte density at the time of sampling ($p=0.473$)". However, the data was adjusted according to gametocyte density? Could the rationale for doing so and its implication to the overall analyses be clearly spelled out in the main text?

10. The selection of proteins was based on two different mass spectrometry datasets conducted at different times by the same authors. Since search algorithms have changed, were the datasets re-analyzed with today's software? Were the datasets compared to other gametocyte proteomic data since each approach can miss protein identifications? The text indicates that schizont proteomic data were used for selection of proteins for the array. The data does not appear to be included in this manuscript, as it should, since it is difficult to assess the data quality. The data should also be uploaded to a recognized database repository and meet MIAPE guidelines.

11. "In total, 16 proteins (30 target peptides) were primarily expressed by asexual stage parasites, 28 (21 target peptides) by sporozoites, 18 (26 target peptides) by immature gametocytes, 95 (153 target peptides) by gametocytes/gametes. 156 (266 target peptides) were shared across stages (though upregulated in mature gametocytes, according to our mass spectrometry analyses), and 13 (20 target peptides) were of unknown stage expression". The method and rationale for the statistical approach for comparing the relative abundance or "upregulation" across different datasets and stages is missing in the manuscript. How were sporozoites included if the selection of proteins were based on trophozoite and gametocyte proteome datasets? Were these sporozoite proteins checked against other published data?

12. In the selection of proteins, the authors argue that protein candidates "must be expressed during gametocyte development or circulation to elicit an antibody response". The latter argument makes sense, but gametocyte development in the bone marrow has not been fully discussed anywhere in the literature. An explanation of how targets present in the bone marrow are relevant for TRA in the mosquito is missing.

13. It is difficult to follow the chain of candidates in the text. As read, it seems that there are 315 targets \diamond 60 targets \diamond 21 targets \diamond 19 targets then back to 62 targets, which turn out to be only 42 unique proteins but some proteins are divided into fragments, each with differing outcomes. Taken into consideration there is no validation for any of these proteins or further sub-categorization into gametocyte/gamete hits, it is difficult to appreciate what is the make-up of the final rank-prioritized list.

14. The reference to peptides instead of protein is confusing in the text. If the peptides are all against a few proteins, then state as much, not "20 novel peptides showed significantly higher seroprevalence".

15. Is it correct that the antibody depletion experiments using recombinant proteins and column chromatography was re-confirmed by ELISA using the same recombinant proteins? It is confusing how this will determine the presence of or control for other reactive antibodies that see different epitopes.

16. SMFAs were done in duplicate but the assay was done two different ways? Why is there a preferred read out? For consistency, shouldn't the read out be the same? It is not clear how two pools of 10 mosquitoes (using the GFP-luc strain) equivalent to assessing (NF54) oocyst counts in a single individual? Given the variation in the SMFA (as well as DMFA), it is unclear why the authors elected to do only two replicate experiments. There are instances where even 3 replicates are not enough. No technical replicates were performed. The statement that such differences make no difference (referencing their own work) to the integrity of the analyses is debatable. Robustness requires consistency and replication (both technical and biological). This seems to be missing here and borne out in Supplemental Table 6 when triplicate SMFAs were actually performed.
17. The authors used 30 kDa filters to remove cytokines or reactive species but no other validation or check was done?
18. Figure 1: Are TR activity < 0% indicative of transmission enhancement? If not, what does it represent? Why were the data not parsed out according to age? Instead of intensity, why not quantify antigen specific IgG?
19. Figure 2: Antibody breadth is higher in 0-5 year olds, so these children are less infectious to mosquitoes?
20. Figure 3: The peptides corresponding to the respective protein is not given anywhere in the manuscript with reference to ≥ 14 of the 61 to ≤ 14 of the 61 targets (or peptides? or proteins?). Which 14? Same 14? <10/20 TR peptides? Are they part of the > 14? How do they partition? Titer of anti P48/45 and/or P230 antibodies with each combination of peptide targets? Protein targets? The information could be conveyed more effectively.

Major/Minor comments to further improve clarity:

- What are the benchmarks that would lead the community to successfully achieve the following goal? Line 22-24, p. 2, introduction: "Identifying the factors responsible for immune TR activity could lead to the development of novel malaria transmission blocking vaccines, and provide useful markers of infectiousness."
- What is the relevance of transmission-reducing activity to malaria transmission? Mosquitoes remain infectious if only a single oocyst remains productive.
- The Table 1 legend suggests that not all samples have available data, yet the text on p. 3 indicates that malariometric data accompanied all plasma samples? Unclear.
- Table 3 requires additional legends to explain column headings and predictions based on what software or source.
- Clearly indicate stage V gametocytes, if this is what the authors want to convey. "Mature gametocytes" have not been clearly defined. Additional text is needed to explain the selection of non-stage V gametocyte proteins for inclusion in the array.
- It is difficult to follow the origin of samples and how they partitioned during analysis/experimentation. "Individuals from malaria endemic areas were all asymptomatic when sampled, and were either recruited randomly (n=42), or based on the observation of malaria parasites (n=273) or specifically gametocytes (n=276) in blood smears". For example, the 42 samples were from which country and what age group? A table summarizing this information would be helpful. Presumably the samples were collected at various periods during a single or several transmission seasons. Since sampling early will likely lead to a different exposure to gametocytes than later in the transmission season, how was this factor controlled in the analyses?

- Of the 6 TRA sera, only 3 had >10 mL. What sort of bias was introduced into the study here?
- “Though our samples were not collected from a random survey of a single population, the prevalence and magnitude of antibody responses to Pfs48/45 (Pfs48/45-10C68) and Pfs230 (230CMB31) increased with age, and this increase was significant after adjustment for gametocyte density (figure 20 1 A & E)”. So the increase was not significant unless the analysis was adjusted for gametocyte density. What is the rationale for adjusting gametocyte density if the breadth of responses and ≥90% TRA did not correlate with higher gametocyte density? This was not very clear. Are gametocyte-negative samples (if really negative) with TRA indicative of prior exposure to gametocytes? If so, the rationale (which is hard to grasp for biologists or non-modelers) for adjusting gametocyte density becomes even more unclear.
- In the text, the TRA analyses as described were for either P230 or P48/45, but not both. Why not both?
- What was the rationale for preparing a 9X concentration of IgG?
- The authors discussed the issue of potentially missing antibodies to P48/45 and P230 that would otherwise not bind to their protein fragments (i.e., not presenting the native epitopes). The ELISA used the same proteins though instead of native antigen or the full-length protein. It is therefore not clear if the quality control was stringent enough.
- How was false discovery calculated before it was used to adjust the analyses?
- The success of anti-P48/45 and P230 antibody purification methods (e.g., coupling efficiency) in this manuscript were based on rabbit and rat IgG affinity, which is different from human antibodies. Not sure what the authors want to convey with the inclusion of the supplemental table 2.
- The exact concentration of the antibodies (following concentration) is not provided. What the exact antigen-specific IgG concentration used in the assays? This would be helpful in understanding what 9X concentration means.
- From supplemental figure 2: “(OR were therefore generated with reference (ref.) to mosquitoes feeding on individuals with responses in the first quartile)”. What is (ref.)?

Minor comments (general):

- Errors in spelling, missing articles or words and poor sentence construction abound. Please correct.
- Malaria is a disease so “malaria life-stage” is nonsensical. The authors probably mean “malaria parasite” or “malarial parasite”.
- Be consistent-- “transmission-blocking” or transmission blocking? “TBI” was used on p. 14. Transmission-blocking immunity? Not defined at first use.

Reviewer #2 (Remarks to the Author):

Malaria transmission-blocking vaccines (TBVs) have been regarded as one of the essential strategies towards malaria elimination. However, there have been only a few TBV candidates under development (Pfs25, Pfs48/45, Pfs230). Therefore there is an urgent need

for screening to identify novel candidate antigens for timely TBV development. To this end, the Authors determined the transmission-reducing (TR) activity of naturally acquired antibodies from 648 malaria-exposed individuals by *Anopheles stephensi* mosquitoes in the standard membrane-feeding assay (SMFA). Immuno-affinity purified α -Pfs48/45 and α -Pfs230 antibodies from a subset of individuals were able to inhibit transmission to mosquitoes, but IgG depleted of these antibodies continued to reduce transmission. To identify responsible antigens to the TR activities, the samples were probed on a microarray comprising 319 gametocyte proteins. Finally, they found out that the TR activity was significantly associated with responses to 41 novel proteins. Additionally, in the direct membrane-feeding assay (DMFA), they found out that *Anopheles gambiae* mosquitoes were significantly less likely to become infected, and at a reduced rate, if feeding on natural gametocyte carriers who were reactive to Pfs48/45, Pfs230, or to separate combinations of newly identified TR associated proteins. I think that the results of this systematic, large-scale and comprehensive study will provide TBV community very important information for developing effective TBVs based on the selected antigen(s). All the works were carefully designed, clearly presented, and the manuscript is well written. I have the following comments for the improvement of this manuscript.

Major comments:

1) Novel gametocyte antigens associated with TR efficacy

The Authors identified an association between antibody responses to 41 newly described proteins and naturally acquired immune TR activity (Table 3). I am sure that this is the most exciting data in addition to the Pfs48/45 & Pfs230 (well-known TBV vaccine antigens) results, however there is no validation results presented. The Authors themselves claimed that this is not in the scope of this study (future work). I understand how much more efforts are needed to produce such validation results. Even though I still think it is very important and attractive to the readers of this journal that animal antibody against at least one novel gametocyte antigen out of 41 shows the TR efficacy in SMFA.

2) Affinity purification of antibodies against Pfs48/45 & Pfs230

<Page 5, Lines 20-22> Results section:

"For all six individuals the α -Pfs48/45 and α -Pfs230 depleted IgG (fraction 5, containing no antibodies against Pfs48-45 epitopes 1-3, or Pfs230 epitope C, confirmed by ELISA) inhibited mosquito transmission. This activity was complement-independent."

<Page 9, Lines 10-14> Discussion section:

"It is highly unlikely that α -Pfs48/45 antibodies to epitopes other than epitopes 1-3 would have contributed considerably to TR activity. Because the activity of α -Pfs230 antibodies is complement dependent, it is similarly unlikely that the TR activity we observed in α -Pfs48/45 and α -Pfs230 antibody depleted IgG fractions is due to antibodies to Pfs230 epitopes outside the region captured by the 230CMB product (aa. 444-730)."

Based on the above description, the Authors emphasized that the completeness of the removal of antibodies against both Pfs48/45 R0-10C and Pfs230CMB. As they know there is no proof that the folding of these cysteine-rich recombinant proteins is identical to the native proteins. Moreover, there is no proof that anti-Pfs230 antibody against outside of Pfs230CMB region still remained after the affinity purification by the recombinant proteins. To clarify this uncertainty, I strongly recommend the Authors to measure anti-Pfs48/45 and

anti-Pfs230 antibodies in six total IgG samples after the affinity purification (Table 2) by the ELISA using *P. falciparum* gamete extract and monoclonal antibodies. Because the Authors are the establishers of this excellent ELISA system. (Ref. Bousema, T. et al. The Dynamics of Naturally Acquired Immune Responses to Plasmodium falciparum Sexual Stage Antigens Pfs230 & Pfs48/45 in a Low Endemic Area in Tanzania. PLoS ONE 5, e14114 (2010).; Ouedraogo, et al. Naturally Acquired Immune Responses to Plasmodium falciparum Sexual Stage Antigens Pfs48/45 and Pfs230 in an Area of Seasonal Transmission. Infect Immun 79, 4957-4964 (2011).)

Minor comments:

1) <Page 6, Lines 24-30> Results section:

"Thirty-four of the 42 proteins....Six were primarily expressed by asexual stages; one by sporozoites; two by immature gametocytes; 12 by gametocytes/gametes; and 21 were shared more equally between multiple life stages....."

Please add this information in the current Table 3. This modification will be helpful to understand easily.

2) Figure 1 D&H

Please add (Figure 1 D&H) in the text.

3) Table 2

"Age/Sex" should be "Sex/Age".

4) Page 12, Line 3

"longer than 15 tears" should be "longer than 15 years".

Reviewer #3 (Remarks to the Author):

I will restrict my comments to the statistical analyses. These analyses are mostly well done but are not always entirely explained.

I have some questions about Figures 1A and 1E. The p-values are said to be "from logistic regression", but this doesn't tell us what hypothesis has been tested or what specific test has been used. Has a statistical test been done to detect differences in average seroprevalence between any of the three age groups (that would be a test on 2 df)? Has a likelihood ratio test been used perhaps? Analogous questions could be asked about Figures 1B and 1F where the p-values are from linear regression.

What do the error bars represent in Figures 1A and 1E? Are they standard errors? Or standard deviations? Or CIs? Do the error bars account for gametocyte density? What are the sample sizes in the three age groups?

I notice that the study participants from The Netherlands are all 26 yrs or older, whereas the subjects from Gambia and Cameroon are all young. Hence the apparent age effects in Figures 1A-B, 1E-F are confounded with possible geographic differences between The

Netherlands vs Gambia and Cameroon. Have the authors checked that the apparent age effects are not just differences between populations?

The linear relationships in Figures 1D and 1H don't look very convincing. This seems partly because of the scale used to measure TR activity. I am guessing that TR activity is measured as $\max(0, (O_c - O_t) / O_c)$ where O_t and O_c are the oocyst numbers in test and control mosquitoes respectively. This measurement scale has the problem of compression of values around zero. Would an alternative, less constrained, TR measure such as $\log(O_t / O_c)$ be more suitable for linear correlation analyses?

The manuscript refers to the p-value adjustment as "adjustment for false discovery", which doesn't quite make sense. A more correct way to describe this process would be say that "p-values were adjusted by the Benjamini and Hochberg method to control the false discovery rate below 5%". The adjusted p-values can be referred to as "FDR" values or as "q-values".

Rather than simply saying that statistical analysis was conducted using R, it would be more usual to cite the major R packages that were used. For the microarray analysis, it seems that the limma package as been used.

The microarray analysis seems basically sound, but the background correction step, with negative values converted to 1, is rather crude. The limma package provides automatic processing of GenePix files and much better background correction methods than used here. The normexp background correction method is one that might well be tried.

Reviewer #1 (Remarks to the Author):

Stone, et al.-- "Unravelling the immune signature of Plasmodium falciparum transmission reducing immunity"

The authors describe a large body of data that "demonstrates that the efficiency of malaria transmission from humans to mosquitoes is modulated by host antibody responses to gametocyte antigens", not only to the P230 and P48/45 but also to a larger repertoire of novel parasite targets. Although at first read this is not a new concept, as the association has been described since the late 90's, the authors now provide the missing evidence supporting a causal association of anti-P230 or anti-P48/45. This is interesting and partially ironic since P230 and P48/45 have already moved forward on several occasions towards vaccine development. It was assumed that there was enough causal rationale to support vaccine development by now. What is really "new" is the discovery of humoral responses to additional Plasmodium proteins. However, not only does the clarity of the manuscript require improvement, but very simple validation experiments are missing to support the discovery.

The methods and analytical approaches, as described, appear a bit flawed in design, but perhaps the rationale is simply missing? The manuscript describes some potentially exciting gems as a result of their work, but a compelling case is still needed. Overall, the manuscript details a comprehensive and almost thorough set of experiments, results and analyses. The addition of a few experiments and key information to "fill in the blanks" for a broader audience is required before further consideration for publication.

We thank the reviewer for their compliments about the comprehensiveness and value of our experiments and results. We further appreciate the reviewer's considerable and detailed feedback. Based on his/her comments, it is evident that our rationale was not clear in the original manuscript, and that significant adjustment to content and flow in line with a redefined rationale, as well as some additional experiments, are required to make improve the manuscripts merits and readability.

As indicated by the reviewer, there is ample evidence supporting the development of Pfs230 and Pfs48/45 as candidate transmission blocking vaccines. Monoclonal antibodies specific to these proteins were shown to inhibit *Plasmodium* transmission in the early 1980's (Rener 1983), proving unequivocally that these proteins are involved in parasite development in the mosquito gut, and that antibodies in the parasites environment can inhibit its development. It has only been assumed that the α -Pfs48/45 and α -Pfs230 antibodies elicited by natural exposure to *Plasmodium* would be similarly functional. In our group we have conducted studies on naturally acquired α -Pfs48/45 and α -Pfs230 antibodies. In the process of publishing the two most recent manuscripts (Jones et al., J Infect. 2015 and Ateba-Ngoa et al Am J Trop Med Hyg. 2016) reviewers highlighted that the statistical associations between the prevalence of α -Pfs48/45 and α -Pfs230 antibodies and TRA should be interpreted with caution and do not demonstrate functionality. Naturally acquired antibodies to these targets may not be as potent as monoclonal antibodies generated in immunisation experiments and might simply reflect the existence of other, functionally more relevant responses. We were therefore surprised by the comment of reviewer 1 on this point. Our analysis is the first to prove the functionality of naturally acquired α -Pfs48/45 and α -Pfs230 antibodies beyond doubt. It has been noted in many studies that individuals with antibodies capable of blocking transmission to mosquitoes did not possess significant antibody titres to either Pfs48/45 or Pfs230. We aimed to describe these antibodies involvement in natural TRA, to move past simply stating the absence of these responses in individuals with TRA as a rationale for seeking new candidates, and to identify antibody responses to proteins other than Pfs48/45 and Pfs230 that might be similarly associated with TR immunity. The latter goal was intended to bring additional antibody responses to light that that are biomarkers of the TRA. We did not intend to indicate that

the correlates of TR immunity we identified were all probable targets of TR antibodies. This was clearly not well communicated in the original manuscript and has now been improved.

To simplify and better define our rationale, we have made significant changes to our introduction and have entirely re-written our abstract, results and discussion sections. The results section is now structured as follows:

- 1. Antibody responses to gametocyte antigens**
 - Assessing epidemiological patterns in antibody responses to known and unknown gametocyte proteins.
- 2. Naturally acquired transmission reducing activity is associated with antibody responses to gametocyte antigens**
 - Assessing the relationship between antibody responses to known and unknown gametocyte proteins, and antibody mediated TRA assessed in the standard membrane feeding assay (SMFA).
- 3. Gametocyte antibody repertoire is associated with transmission efficiency during natural infections**
 - Assessing the relationship between antibody responses to known and unknown gametocyte proteins, and gametocyte transmission to locally relevant mosquitoes during naturally acquired infections assessed in the direct membrane feeding assay (DMFA).
- 4. Evidence for a functional role of specific antibody responses in naturally acquired transmission-reducing activity**
 - Experiments with affinity purified antibodies to characterised proteins Pfs48/45 and Pfs230 to assess their functional importance in TRA
 - Identifying plausible transmission-inhibiting antibody targets among all TRA biomarkers
 - Assessing the relationship between antibody responses to these targets and gametocyte transmission to locally relevant mosquitoes during naturally acquired infections.
 - Experimental validation of one marker, PfGEST, using a panel of mAbs
 - Surface immune-fluorescence assays with WT and Pfs48/45 KO gametes, using Pfs48/45, Pfs230 and PfGEST mAb, and total IgG and α -Pfs48/45 and α -Pfs230 Ab depleted sera.

Based on the comments from reviewers 1 and 2, we also performed considerable additional experimental data collection, as well as additional bioinformatics and statistical analyses to better describe both the array protein characteristics and associations with TRA and donor characteristics. Major changes to the results are as follows:

- A clear distinction has been made between the identification of TRA biomarkers, which was the endpoint of our previous manuscript, and the identification of biomarkers that should be prioritised for functional validation as being possibly involved in gamete viability (and thus potential targets for neutralisation by antibodies).
- As part of this distinction, more detail is provided for all antigens that were printed on the array, using a bioinformatics approach that utilizes all currently available gametocyte proteomic datasets.
- To determine the presence of naturally-acquired antibodies binding to unknown gamete surface antigens, we performed surface immuno-fluorescence assays (SIFA) using gametes of a Pfs48/45 knock-out (KO) parasite (van Dijk, Cell, 2001) that lacks both Pfs48/45 and surface retained Pfs230 (Eksi, Molecular Microbiology, 2006). Pfs48/45 and Pfs230 specific mAbs confirmed the absence of these proteins on Pfs48/45 KO gamete surfaces. The immuno-fluorescence of gamete surfaces observed using total IgG and IgG depleted of α -Pfs48/45 and α -Pfs230 antibodies from naturally exposed

individuals therefore reflects recognition of unknown antigens on the gamete surface. These data have been compiled into a new photographic figure – figure 4.

- Monoclonal antibodies against one of the antigens statistically associated with TRA and plausibly mechanistically involved in TRA have been tested in duplicate dose-response experiments the SMFA.

Major comments:

1. It is difficult to understand why antibodies that were reactive with a subset of the novel proteins were not tested for IFA staining of stage V gametocytes or gametes following induction of gametogenesis. The authors clearly have all the reagents in hand and it would go a long way to providing the data to show “relevance” of some of the presumed targets identified by the protein array.

The reviewer suggested that we should determine whether proteins identified in our analysis as being statistically correlated with TR immunity are present on the gametocyte/gamete surface, which would make the proteins viable targets of antibody binding in the mosquito midgut. This would indeed go a long way toward bridging the gap between the statistical associations we have shown, and a plausible role in TR immunity. Unfortunately we do not, as the reviewer suggests, have all reagents in hand to achieve this. Proving the presence of any specific novel proteins on the gamete surface with IFAs requires protein specific antibodies; our experiments used immune sera as antibody sources, with varying degrees of antibody recognition. Generating the antibodies would require additional funding, ethical approval, production of the proteins at scale and ideally in their native conformation (microarray production requires comparatively negligible quantities of protein; immunisation considerably more), rodent immunisations, before IFAs and SMFAs could be considered. The production of gametocyte antigens has proven complicated, illustrated by the fact that it has taken more than a decade to produce recombinant Pfs230 and Pfs48/45 in different expression systems to produce sufficient quantities of conformational protein to allow immunisation experiments. Whilst the same level of dependency on conformational protein may not be required for the proposed experiments, even in an optimistic scenario it may take over a year to obtain the reagents required for the proposed validation of surface expression for all 43 novel proteins.

To address the reviewer comments, we performed the proposed experiments on surface expression and functionality for one of the novel proteins for which we were able to source a panel of monoclonal antibodies. Experiments for the remaining 42 proteins (for which no products for immunisation of monoclonal antibodies are available in the foreseeable future) are beyond the scope of the current manuscript. We do acknowledge that the original manuscript lacked clarity on this point and was perhaps misleading in the sense that all ‘hits’ were presented as possible surface antigens.

In the revised manuscript, we have thus made clearer that we are not aiming to functionally validate all our TRA biomarkers as potential TBV candidates and some candidates can be disqualified as promising targets based on existing literature.

Specifically, we have taken a number of steps with these comments in mind.

1. We have redefined our approach, making clearer that we did not anticipate that all TRA associated antibody responses were mechanistically linked with TRA, and that functional validation was not our aim. Rather, we aimed to reveal biomarkers, and enable the prioritisation of candidates for functional validation.
2. We have obtained monoclonal antibodies specific to what we consider a promising TBV candidates on our list: Gamete egress and sporozoite traversal protein (GEST). The production of this antibody was part of a separate collaboration between the PATH malaria vaccine initiative, and the Hypothesis-driven Pre-erythrocytic Antigen Target Identification Consortium (HPATIC).

We also produced *P. berghei* parasites in which GEST was separately knocked-out, and GST tagged. Fluorescence microscopy indicated that the protein was not present on the gametocyte surface, confirming previous data indicating that this protein was exported through the membrane and involved in gamete egress. Importantly, our results also confirmed that GEST knockouts had impaired oocyst production. These results are not included in the current manuscript, as similar data has been presented for PbGEST knockouts previously, providing a more comprehensive reference for our focus on this proteins validation (Talman, Molecular Microbiology, 2011:) but can be made available for the reviewer if requested. We have included our SIFA experiments with PfGEST antibodies and *P. falciparum* gametes (demonstrating no clear surface recognition) and SMFA experiments (demonstrating very limited and variable TRA).

3. We have performed IFAs using a selection of sera with high TRA. Though these experiments do not prove the presence of our TRA biomarkers on the gamete surface, they do indicate that immune sera recognise antigens on the surface of gametes other than Pfs230 and Pfs48/45. This provides an experimental validation of our microarray based approach to discovering new gametocyte immune responses, and our analysis of their association with TRA.

As it stands the list of novel candidates remains long and the rationale for their selection in the first place is missing. The authors should have prioritized the analysis of a subset of targets, especially those that are predicted NOT to be on the cell surface of gametocytes or gametes. It is odd that “presence of a transmembrane membrane domain” was not considered as a potential red-herring, since there are several proteins that are not secreted to the cell surface that have such a domain signature, e.g., ER and Golgi proteins, as obvious examples. Inner membrane proteins can also fall into this category. The authors included 19/42 proteins in Table 3 that are likely not to be the targets of antibody-mediated transmission-reducing activity as they are not on the cell surface. Mitochondrial, nuclear and apicoplast proteins have been included? Are the authors suggesting that a new blocking mechanism is in effect? Also, the inclusion of proteins in the list despite not meeting their selection criteria is puzzling. If there is a null prediction for any relevance to the proposed biological mechanism of antibody-mediated inhibition, then why include it in the list in a main table? It is as if the list was not checked again. For example, randomly selecting PF3D7_0505000 identifies a protein with no predicted location (according to Table 3) but the potential presence of 4 transmembrane domains (supplemental Table 4) likely keep it in the list (though this is a soft inclusion-criterion). Yet, the fact that it has a Dos2-interacting transcription regulator of RNA-Pol-II domain (PlasmoDB) should have potentially precluded it from the main table. It is recognized that the authors would like to demonstrate associations between transmission-reducing activity and antibody recognition of a protein following the probing of a protein array, but some caution is merited here and a closer curation of the data is needed (with the accompanying plausible rationale). Including a table of data highlighting associations with potentially limited biological relevance should be avoided.

We thank the reviewer for these very important comments that have prompted us to rigorously revise of data presentation. It was not our aim to present all identified proteins as potential vaccine candidates and we agree that caution is needed here in order not to mislead readers and stimulate research on proteins that are unlikely to be of functional importance.

As the reviewer pointed out, our inclusion criteria for the array proteins does not mark them out as surface proteins – no characteristic or group of characteristics can do this with absolute certainty in the absence of IFA data with specific antibody. Instead, we opted to include proteins that were more abundant in gametocytes, which possessed characteristics that would be necessary for surface expression (i.e. we chose to be inclusive). It is highly likely that internal, or non-essential surface proteins also elicit antibody responses that associate with TRA, despite having no role in its activity. Our intention when constructing the array was therefore to focus on a protein library that included

most probable transmission reducing antibody targets, but was by necessity not exclusive to them. The reviewer specifically questioned the value of supplemental table 3, which is a list of the proteins included on the array, along with minimal details of these proteins (weight, transmembrane domains, and signal peptides). As these were and remain criteria for the inclusion of the proteins on our array, we think this table is appropriate. We have now provided a more thorough bioinformatics analysis of all considered genes that gives the reader direct access to all relevant information on gametocyte specificity of genes and plausible surface expression. This analysis (see bullet 2 below) was used to prioritize genes that may be of particular interest for functional characterisation. In the revised manuscript, we thus improved clarity on our rationale and description of the array. In line with the reviewer suggestions, we conclude our manuscript with a focus on possible mechanistic involvement of a sub-selection of these TRA biomarkers in transmission inhibition. To this end, we have:

1. Adjusted our manuscript so that discussion of antibodies with potential mechanistic involvement in TRA (theoretical or proven) comes after all statistical/serological investigation, providing a more logical conclusion to the manuscript and avoiding a misinterpretation of biomarkers of TRA that are unlikely to be functionally important.
2. Prioritized a sub-selection of the larger list of TRA associated responses based on the plausibility of their mechanistic involvement in transmission reducing immunity. We have analysed these proteins association with natural gametocyte transmission separately from the other TRA biomarkers, as suggested by the reviewer.

For this categorisation, we utilised the publically available protein annotation data described by the reviewer where these were available, to exclude proteins that were known or predicted to be internal. For proteins without clear GO terms, we categorised proteins based on protein homology using HHPred protein prdomain prediction (MPI bioinformatics Toolkit as an integrative platform for advanced protein sequence and structure analysis, Alva V, Nam SZ, Söding J, Lupas AN. *Nucleic Acids Res.* 2016 Jul 8;44(W1):W410-5). As described, we cannot be certain of gamete surface expression or involvement in gamete viability without experimentation of the type we have described. This categorisation is therefore based on *exclusion* of probable internal proteins, rather than inclusion of probable surface proteins.

2. The authors should be commended on identifying the major drawbacks of the protein array (a major approach in their study). However, these statements are not enough to explain the results nor does it excuse potentially misleading conclusions (point 1). Some quality control of the results in the form of validation experiments is necessary. It is a bit of a disservice to the community if the data are simply put out there as if they are already leading vaccine candidates for consideration. The authors should recognize that they are an influential body to the larger malaria community and the data need to be carefully presented so that others do not waste time and valuable research funds chasing red herrings.

We take this last comment very seriously, and hope that collectively the redefined rationale, separation of results into biomarkers and candidates for further validation, and additional experimental validation, meet with the reviewer's approval.

The specific point the reviewer makes regarding the microarray methodology is however slightly confusing. The drawbacks of the methodology we discuss (i.e. downfalls of the platform itself) are unrelated to the selection of proteins for the array (and therefore the identity of TRA associated responses). We intended to make clear that the downfalls of the array methodology (primarily, improper protein folding) may give rise to a list of correlates that may be an incomplete representation of the proteins eliciting antibody responses associated with TR immunity. We have specifically addressed these disadvantages in our revised discussion section. Pfs48/45 is mentioned as an illustrative example of a protein that is functionally relevant but is not expressed in the correct confirmation on the array. The findings from the array should therefore, as we write, be considered as a 'rule in' rather than 'rule out' approach.

3. What level of stringency was applied in the probing of the protein arrays? No serum dilutions (dose-response curve) were performed. No orthogonal validation using a different assay was performed to confirm the read out? What was the density of proteins or peptides of a given protein per spot? How were the “multiple fragments” or domains of a given protein identified for representation on the array? What was the rationale behind the selection and how was the rationale demonstrated to be correct or reasonable? P48/45 was not included in Table 3. Is this not considered a positive control for the assay? The array described in the manuscript appears to be new and therefore the same level of measurement to ensure rigor and robustness needs to be applied.

The reviewer is correct that serum was probed on microarray only at physiological concentration. Microarray analyses on this scale are expensive (in the order of 200 USD per sample for this custom-made array), and serum dilutions of all samples would have required manifold increases in the number of arrays we produced. We feel that the added value of serum dilutions of a handful of samples would be limited. Rather, we chose to probe a larger number of samples with known SMFA and DMFA outcomes to assess concordance of hits in two independent and distinct transmission experiments. We believe that the custom made array used on a large sample set that is truly unique in terms of detail on TRA in SMFA and DMFA is the main strength of this manuscript.

The reviewer noted that we were clear in our manuscript about the downfalls of the microarray technology. One additional factor we could have discussed specifically is the issue of protein concentration. In the IVTT protein production system used by Antigen Discovery Inc., protein concentration is indeed unknown. Quality control assures acceptable protein abundance, as indicated by positive HIS and HA tagging and visualization. This is a well-known limitation of this system, and restricts the interpretation of the data to comparisons of signal intensity between the same proteins by different individuals, rather than between different protein spots. We have been careful to limit our analyses with this in mind. Similarly, the signal intensity is not, and should not be interpreted as an antibody titre.

Details of the array protein production system specifically mentioned by the reviewer have been added to the methods section:

The microarray proteins were produced in an Escherichia coli based in vitro transcription-translation (IVTT) system. As conformation is not verified for all proteins, those requiring post-translational modifications will likely not have been expressed in their native conformation. This was exemplified by Pfs48/45 that has a known complex conformational structure^{22,53} and showed minimal reactivity or discriminative power when printed on the array as an IVTT product (supplemental table 3). This apparent disadvantage of the IVTT system will plausibly have affected our results for other conformational proteins, so the list of proteins in table 3 must be treated as a ‘rule in’ rather than ‘rule out’ selection for proteins associated with antibody mediated TRA.

One protein (Pfs230 CMB) was used in ELISA and was printed directly onto the microarray, providing an opportunity for orthogonal validation as the reviewer suggests. We have added the following to the data analysis section of the methods:

Quality control plots were made after each treatment of the microarray data, which included boxplots and density plots of probe intensity by study sample and probe type. Recombinant Pfs230 CMB was printed on the array for comparison to ELISA with the same protein, yielding a correlation co-efficient of 0.44 ($p < 0.001$).

The use of IVTT products mapping to Pfs48/45 and Pfs230 as positive controls for the array (in terms of their association with TRA) is unfortunately not possible. The transmission blocking epitopes of Pfs48/45 are conformational, and only elicit TBI upon immunisation if produced with disulphide-bond catalysts in their complex tertiary structure. Accordingly, the IVTT peptide on our array corresponding to Pfs48/45 showed no association with TRA. Pfs230 is large and complex, but the C-

region that is the primary TBV candidate elicits transmission-blocking antibodies in standard in vitro expression systems. Peptides corresponding to Pfs230 on our array were borderline significantly associated with TRA. In fact, in the original manuscript, this peptide was significantly associated; the association became non-significant after the data correction suggested by reviewer 3. The full length Pfs230-CMB protein printed on the array (as indicated above) however showed a strong association with TRA, comparable to the ELISA data (OR 8.7, 95% CI 3.4-22.4, $p < 0.001$). Because we had correctly folded products available for these proteins, we used the ELISA data for all associations with TRA. We have been very clear in the manuscript that the simplicity of the array protein production system means that our list of TRA associated proteins may be missing proteins associated or causal to TRA if they had been produced in a conformation nearer to their native one. We have also now explicitly referenced this issue for Pfs48/45 in the discussion as indicated above.

4. It seems that not all of the DMFAs were performed in the same manner. It is assumed that the whole blood and serum-replacement experiments are all from the same donor. If this is not the case across all endemic sites that performed the DMFA, then this should be made clearer to the reader. Also, did all the sites performing DMFAs use the same operational cut-off for enrolment of an individual in the study in terms of gametocyte density or age? Did they all use the same mosquitoes? In clarifying these sections, the following question should be addressed: “would whole blood DMFAs change the prioritization of the 41 proteins?”

The availability of the DMFA data in these studies provided a unique opportunity to compare and validate the results of the SMFA, and investigate the infectivity of individuals with antibody profiles identified as being correlated with TRA. In our study though, DMFA was not used for the identification of responses associated with TRA. The immune profile associated with TRA was defined based on the SMFA only. Serum replacement experiments could theoretically be used to determine the contribution of serum TR factors to patient infectivity, by comparing the infectivity of reconstituted whole blood and blood in which the serum has been replaced with naïve serum (lacking the patient’s own immune factors). This would have been an interesting alternative approach to assessing TR immunity but has several disadvantages compared to SMFA. The procedure of serum replacement leads to a delay in offering the blood meal to mosquitoes that can result in a significant loss (~40%) in transmission efficiency (Bousema et al., PLoS ONE 2012) and other serum components such as drug concentrations may also influence transmission efficiency and thus reduce both accuracy and precision of serum replacement-DMFA experiments. Importantly, the DMFA experiments are commonly single experiments with considerable noise whilst the SMFA was performed repeatedly. Practically, serum replacement was not performed for the majority of field samples, so analysing this data in depth would have affected our study power. Thus, only infectivity data from whole blood samples were used (representing absolute infectivity with minimal sample handling prior to feeding and generally considered the most robust DMFA outcomes). The transmission efficiency in these DMFA experiments (expressed as % of infectious feeds and % of infected mosquitoes) was related to the donor’s immune profile, using the immune signature of TRA as identified in the SMFA. In this way we used the most robust assay for assessing TRA (the SMFA) to define the relevant immune profile that was then examined in the experiments that are most representative for natural transmission efficiency (the DMFA).

As the reviewer correctly points out, the conditions of the DMFA did differ slightly between locations. However, our language may have been misleading regarding the extent of these differences. In all cases, the patient’s own cellular content was fed to *A. gambiae (s.l)* mosquitoes in the presence of the patients own serum using the same feeder system and identical glass feeders that were all sourced through Radboud university medical center. We have improved the language in our methods section to avoid confusion regarding the DMFA, and have made clearer in the manuscript that all sites used locally caught and reared colonies of mosquitoes, as indicated.

DMFA was performed during the original studies for 498/579 individuals native of endemic areas, as described in the separate study protocols using identical glass mini-feeders⁷¹ (Coelen

*Glastechniek, Weldaad, the Netherlands) and the same experimental protocol⁹⁶. Of the 498 individuals with DMFA data, 494 had microscopy based estimates of asexual and gametocyte density; only gametocyte positives were used in our DMFA analysis. All sites used local colonies of *A. gambiae* s.s. (>90%) (The Gambia), *A. coluzzii* (Cameroon and Bobo Dioulasso in Burkina Faso) or a colony comprising a mixture of *A. coluzzii*, *A. gambiae* s.s. and hybrid forms (other Burkina Faso sites). Data presented in the current manuscript are the proportion of infectious individuals and percent of mosquitoes infected with oocysts after feeding on whole blood 96.*

The issues of age and gametocyte density touched upon here are addressed in our response to point 5 by reviewer 1, and in our response to comments by reviewer 3.

To additionally address the concerns of the reviewer, we now perform all DMFA analysis on individuals with confirmed gametocytaemia by microscopy. Though the results change very little, we feel that restricting the analysis to this group increases the robustness of our conclusions regarding the association of antibody responses and infectivity. These analysis are still adjusted for differences in gametocyte density between groups, as described here.

5. What does “All the following analyses are adjusted for gametocyte density at the moment of DMFA” mean? As written, these appear to be “convenience samples”, not a study done specifically for the project described in this manuscript, with the exception of a few from Cameroon. What was the gametocyte density (operational definition) at the time of the DMFA in each site?

The reviewer is correct that these were samples gathered from various previous studies that were all designed to examine transmission efficiency but not specifically to unravel the immune signature of TRA. The use of a large sample set from different endemic regions was essential because TRA is very uncommon, particularly with the stringent thresholds we used for defining our ‘high-level’ activity (>90% reduction in oocyst intensity in the SMFA). We view the range of sites we were able to gather samples from as an advantage – the same number of samples from a single site may yield less generalizable results. The studies from which samples were derived are referenced in the methods and in Table 1. Table 1 also gives the parasite and gametocyte prevalence across the various sites.

Gametocyte density by microscopy at the time of sampling is the only direct measure we have of gametocyte exposure. As TRA is closely linked with exposure to gametocyte antigens, we felt that associations of immune response and TRA should include gametocyte density as a potential confounder. Rather than writing ‘after adjustment for gametocyte density’ after each OR and p-value in our results section, we simply chose to state that all analysis were multi-variate, and included gametocyte density as a potential confounder. This has been made clearer in the results section in the revised manuscript. We have now also included gametocyte density (median (IQR)) for each site in the methods section, along with detail of the density calculation (in response to point 6 by reviewer 1).

6. What was the baseline WBC count used for all microscopic evaluations? The published literature suggests that different sites use different standards for accuracy in counting parasites in blood smears. At what point is a slide considered gametocyte-negative? Is it counted against 200 WBC or 1000 WBC? How do such differences, if they are indeed performed differently, going to affect the overall analyses?

To address the reviewer’s general concern, we have added detail to the methods section to make clearer how microscopy was performed at different study locations.

As the reviewer correctly points out, the samples we used are from multiple surveys with varying inclusion criteria. This was essential for the reasons outlined above – we simply could not have performed an analysis of high-level TR immunity without being very inclusive in our selection criteria.

We did not assess the relationship of array responses and gametocyte positivity by microscopy, for exactly the reasons the reviewer mentions: that some gametocyte negatives by microscopy may be positive for gametocytes by more sensitive methods. The thresholds for defining positivity differed between studies but we consider it highly unlikely that this affects the validity of our findings since we adopted a conservative approach in the analysis of DMFA data. Individuals with TR immunity are by definition gametocyte exposed. For the assessment of antibody responses associated with TR immunity then, the only valid comparison is to compare the responses of the individuals with TR immunity, to gametocyte exposed individuals without TR immunity (to remove noise relating just to gametocyte exposure). Positivity by microscopy was our only absolute assurance of gametocyte exposure, so this was used to define the comparison group. In analysing the DMFA results, we thus compared transmission outcomes for gametocyte carriers with and without the immune profile associated with TRA in the SMFA, adjusting for the estimated density of gametocytes at the moment of feeding. These associations are unlikely to be affected by the sensitivity of gametocyte screening that is more likely to affect gametocyte prevalence (and thereby the likelihood to be included in the analyses) than density.

Within specific locations or studies, it would be very interesting to use the microarray to assess differences in response based on gametocyte status. Our data, which will be provided in an online repository, may provide a useful starting point for such studies.

7. No human IgG subclass assessment was conducted?

This is an interesting point and good suggestion. There has been limited study into antibody subclass and transmission reducing immunity. Antibody inhibition of gamete fertilisation with Pfs230 antibodies is mediated by the activation of complement, indicating a role for IgG1-3 but excluding a role for IgG4. Because of the scale of our experiments, and because we focused on studying the associations of a battery of antibody responses with the presence of TRA of any sort, focusing on antibody subtype was outside the scope of this paper. This analysis would also have multiplied by several-fold the number of arrays required, which given the large scale of the study was unfortunately impossible. Future studies aiming to uncover the exact mechanism of naturally acquired TRA (not only identifying the presence of TRA associated antibodies as we are here) would be well advised to assess antibody subclass. We highlight in the discussion section that the imperfect association between (total IgG) antibody titre and TRA suggests that future experiments may focus on antibody functionality, which may include a more detailed description of IgG subclass.

*Our findings identify the further characterisation of functional and non-functional gametocyte antibody responses in individuals with naturally acquired TRA as a research priority. Whilst the prevalence of high-level TRA decreases with age, an opposite trend is observed for the prevalence of antibodies to Pfs48/45 and Pfs230 and to newly identified gametocyte antigens (**supplemental figure 5**). A better characterisation of these antibody responses in relation to functional activity is needed and may include a detailed assessment of antibody isotype, affinity and complement-binding activities.*

8. Approximately 3.3% (22/648) samples had potent TRA. What age group? Which countries? Similar across the board? Is TRA $\geq 80\%$ a meaningful benchmark? As written, it is difficult to follow what constitutes a meaningful biological phenomenon in terms of reduction of oocyst numbers in mosquitoes. The goal, as stated in the text is to find “biomarkers of broader sexual-stage immunity”. Would the small proportion of high blockers in the entire dataset of 648 from different sites experiencing different malaria exposure detrimental to the overall analysis and subsequent conclusions? We have provided the requested details in a table below. The small number of individuals with high-level TRA makes it difficult to address some of the posed questions with certainty. Part of this difficulty was our own stringency in accepting only individuals causing $>90\%$

reductions in oocyst intensity into the final analyses. As indicated in Table 1 (which also details the locations, age ranges and parasitological status of the individuals in our study – as requested by the reviewer), the numbers of individuals with >50% and >80% TRA were considerably greater than the number with >90%. Both 50% and 90% are common benchmarks for TRA in the literature but only 90% has been supported as a replicable level of high TRA in previous studies, notably by van der Kolk and colleagues (Parasitology, 2005).

We chose to use the 80% cut-off to define which individuals would be repeated in the SMFA, and a 90% cut-off to define the primary group for analysis. This is why the proportion of individuals with >80% TRA was specifically mentioned in the text, however, because this information is available in Table 1, we have removed this from the redrafted manuscript in the interests of clarity. We further clarified that we do not consider lower levels of TRA irrelevant. On the contrary, incomplete TRA can have considerable consequences in reducing transmission as shown by Blagborough and colleagues (Nature communications, 2013). However, we defined a strict definition of high level TRA (>90%) based on our extensive experience in assay reproducibility and desire to avoid all sources of uncertainty. We have added the following to the results section for clarity:

Though intermediate TRA may be relevant for transmission⁷⁴ there are uncertainties about its reliable measurement in the SMFA⁷⁵. TRA was therefore dichotomized into ≥90% (high-level TRA) and <10% TRA (no TRA) before assessing associations with antibody responses.

There is extensive discussion about whether TRA is more usefully determined from reductions in the prevalence of infected mosquitoes, or the number of oocysts they are infected with. In this instance, we used the more standard measure based on oocyst number, which is more robust to the unavoidable variations in control oocyst intensity/prevalence that occur within and between feeds using different gametocyte feed material. This approach is supported by analyses from independent groups that repeatedly confirmed that reductions in oocyst intensity are more robust and more reproducible than reductions in oocyst prevalence (Miura, Vaccine 2016) whilst the two are strongly related and any large reduction in oocyst intensity will result in reductions in oocyst prevalence when natural exposure levels are used (Churcher, International journal of parasitology, 2012). Whilst mosquito infection prevalence may thus be a more intuitive biological readout for TRA, oocyst intensity is the most appropriate read out for the SMFA in terms of sensitivity and reproducibility.

Details on the origin of individuals with high-level TRA are provided below. We currently do not consider this table to be essential for data interpretation but are willing to incorporate it in the supporting information if deemed relevant.

Optional supplemental table

ID	Sex	Age	Asexuals/μL	Gametocytes/μL	Location	α-Pfs230 +	α-Pfs48/45 +	TRA %
1	M	69	Infected	35	Netherlands	1	1	100
2	F	11	0	80	Cameroon	1	1	100
3	-	-	-	-	Netherlands	1	1	100
4	M	29	Infected	-	Netherlands	1	1	99.5
5	-	8	9000	5	Gambia	1	0	99.1
6	M	26	-	-	Netherlands	1	1	98.9
7	F	12	-	42	Cameroon	0	0	98.8
8	F	6	-	46	Cameroon	0	0	98.6
9	F	6	2473	15	Cameroon	1	1	98.3
10	-	10	676	20	Gambia	0	1	96.9
11	-	4	5000	65	Gambia	0	0	96.8
12	F	25	0	0	Burkina Faso	1	1	96.6

13	-	5	93500	615	Gambia	0	0	96.2
14	-	5	500	5	Gambia	0	0	96.1
15	M	7	5232	26	Cameroon	0	0	95.7
16	F	5	-	-	Cameroon	0	0	95.7
17	M	4	3960	158	Burkina Faso	0	0	94.7
18	-	7	29000	0	Gambia	1	0	94.0
19	-	-	-	-	Burkina Faso	0	0	92.5
20	-	4	2000	10200	Gambia	0	0	92.3
21	F	33	1017	0	Burkina Faso	1	1	92.2
22	-	74	0	0	Netherlands	1	0	90.7

α -Pfs230/Pfs4845 + Seropositivity
 TRA % Transmission reducing activity %

9. The following is not very clear at initial read. “On a continuous scale, TRA was not associated to gametocyte density at the time of sampling (p=0.473)”. However, the data was adjusted according to gametocyte density? Could the rationale for doing so and its implication to the overall analyses be clearly spelled out in the main text?

This was a test of the relationship between gametocyte density and TRA (simple linear regression) for all individuals. Because there is significant variability in the SMFA at lower levels of TRA (<90%), the reviewer is correct that this analysis is uninformative. Because gametocyte density and gametocyte immunity are causally linked, it is prudent and conservative that later testing of the differences in immune response between sub-selections of individuals (in our primary analysis, individuals with <10% TRA, and individuals with >90% TRA) should be adjusted for differences in gametocyte density between groups, as these may have been causal or confounding.

We consulted the study statistician about the reviewer’s comments, and have made our use of gametocyte density as a potential confounder more clear in the manuscript, as follows:

On a continuous scale, TRA decreased with age, but was not significantly associated with gametocyte density at the point of sampling (p=0.269) (supplemental figure 2). Because previous gametocyte exposure is associated with TRA^{37,72,73}, comparisons between sub-groups were nevertheless adjusted for gametocyte density.

The p value has changed here because: We reanalysed this data based on suggestion by reviewer 3 using 1. Different data correction 2. Simple linear regression, rather than categorical regression, using the log(Ot/Oc).

10. The selection of proteins was based on two different mass spectrometry datasets conducted at different times by the same authors. Since search algorithms have changed, were the datasets re-analysed with today’s software? Were the datasets compared to other gametocyte proteomic data since each approach can miss protein identifications? The text indicates that schizont proteomic data were used for selection of proteins for the array. The data does not appear to be included in this manuscript, as it should, since it is difficult to assess the data quality. The data should also be uploaded to a recognized database repository and meet MIAPE guidelines.

We have provided considerable additional detail in response to this comment. During array construction, we opted to use the best available *P. falciparum* gametocyte proteomic data at the time, generated by Edwin Lasonder at University of Plymouth. Dr Lasonder has more than 15 years of experience in the field of gametocyte proteomics. To generate a metric for gametocyte upregulation we used asexual data generated in the Lasonder laboratory. We judged this to be a more valuable approach than using only the gametocyte data, and analysing it with reference to

asexual proteomic data generated elsewhere. The reviewer is correct that the schizont proteome data was previously unpublished – we have therefore included this data as a supplemental table as requested. The data will also be uploaded to a public repository as requested.

To address the reviewers concerns about our use of data from a single laboratory, we have edited our results and methods sections heavily, to assess the abundance of the proteins included on the array (and those emerging as hits in our downstream analysis) in gametocytes in all previously published gametocyte proteomic datasets. We believe that this is a highly relevant improvement of the manuscript and we thank the reviewer for suggesting a more rigorous analysis of gametocyte specificity. During the period of manuscript revision, we completed a Bayesian integration of 17 publicly-available mass spectrometry and transcriptomic data sets. We used this approach to define a gametocyte specificity score for all genes included on the protein microarray with high confidence. This analysis was published at an online depository in recent weeks (Meerstein-Kessel 2017: <https://www.biorxiv.org/content/biorxiv/early/2017/10/10/199356>) and is under review in a peer-reviewed journal. Though this analysis is only retrospective (and did not inform array construction), we have provided extensive details of the approach taken in our methodology, and provided the raw data in a supplemental table for reviewers (supplemental_table_1_extended_for_review.xlsx). We are confident that this approach, using all currently available data on gametocyte-specific proteins, will be of great value to the reader and will also satisfy the reviewer comments on the need for more detail on the (gametocyte-specific) nature of the proteins on the array.

11. “In total, 16 proteins (30 target peptides) were primarily expressed by asexual stage parasites, 28 (21 target peptides) by sporozoites, 18 (26 target peptides) by immature gametocytes, 95 (153 target peptides) by gametocytes/gametes. 156 (266 target peptides) were shared across stages (though upregulated in mature gametocytes, according to our mass spectrometry analyses), and 13 (20 target peptides) were of unknown stage expression”. The method and rationale for the statistical approach for comparing the relative abundance or “upregulation” across different datasets and stages is missing in the manuscript. How were sporozoites included if the selection of proteins were based on trophozoite and gametocyte proteome datasets? Were these sporozoite proteins checked against other published data?

As we have now made far clearer, gametocyte specificity was not requisite for inclusion on the array – we were interested in biomarkers TRA, and hoped to enrich the selection for more specific and plausibly mechanistic proteins. In fact, numerous non-gametocyte markers were purposefully included on the array as biomarkers of parasite exposure. Even excluding proteins included based on *a priori* criteria, we used a simple set of rules to include other proteins on the array which do not exclude the possibility that they may be expressed by multiple life-stages. Our retrospective analysis of the array proteins in terms of their gametocyte specificity in all published *P. falciparum* gametocyte proteomic studies (described in our response to point 10) addresses this lack of clarity in the original manuscript. We chose not to exclude any proteins prior to analysis in order not to objectively over-power our analysis (by reducing the thresholds for false discovery) and because this wasn't in line with our (now better defined) objectives: only antibody responses related to TRA would be apparent in the output of our analysis. If non-gametocyte specific responses are also related, these may be of interest as non-mechanistic proxies of TRA, indicating that high exposure to other life stages increases the likelihood of being a transmission-blocker. Indeed, this bore out in our analysis.

We have removed this small reference to specific life stage abundance from the manuscript, and have replaced it with an indication of relative gene expression in asexuals and gametocytes, which is given as a summary in table 4 and in more detail supplemental tables. If the reviewer/editor would like us include broader predictions of specific life stage abundance, this can be done upon request.

The sporozoite protein itself was included on the array because it is apparent in some gametocyte proteomic databases (Florens 2002), and because it is a well-known surface protein present

primarily in late mosquito stage parasites. It was included speculatively, and appears to be a correlate of TRA. Further experimental validation is required to rule out its mechanistic involvement in TRA, which we accept – would be surprising.

12. In the selection of proteins, the authors argue that protein candidates “must be expressed during gametocyte development or circulation to elicit an antibody response”. The latter argument makes sense, but gametocyte development in the bone marrow has not been fully discussed anywhere in the literature. An explanation of how targets present in the bone marrow are relevant for TRA in the mosquito is missing.

Our wording was misleading here. In fact, we are co-authors on a report recently submitted that adds significant weight to the evidence that early gametocyte proteins (possibly involved in gametocyte sequestration) may be targets of TR antibodies (Dantzer *et al*). These may ultimately reduce the number of gametocytes that enter the circulation in their maturity. As there is limited evidence in the literature currently supporting this theory (Saeed *et al.* 2008, Sutherland *et al.* 2008, Dinko *et al.* 2016), we agree that is better in this instance to focus on the TR immunity arising from gametocyte/gamete protein responses, active in the mosquito gut. We have edited the wording here accordingly.

13. It is difficult to follow the chain of candidates in the text. As read, it seems that there are 315 targets◊ 60 targets◊ 21 targets◊ 19 targets then back to 62 targets, which turn out to be only 42 unique proteins but some proteins are divided into fragments, each with differing outcomes. Taken into consideration there is no validation for any of these proteins or further sub-categorization into gametocyte/gamete hits, it is difficult to appreciate what is the make-up of the final rank-prioritized list.

14. The reference to peptides instead of protein is confusing in the text. If the peptides are all against a few proteins, then state as much, not “20 novel peptides showed significantly higher seroprevalence”.

This is a highly relevant comment; the lack of clarity obviously coming from proteins that are expressed as multiple protein spots on the array. We now use the term microarray proteins or targets when referring to array spots (the principle unit for our analysis), and protein to refer to the native products these targets map to. We have also made better distinction of which units are analysed. The TRA biomarkers are now also split up into two tables – one presenting all significant microarray targets, the other presenting the proteins they map to and details of these proteins.

15. Is it correct that the antibody depletion experiments using recombinant proteins and column chromatography was re-confirmed by ELISA using the same recombinant proteins? It is confusing how this will determine the presence of or control for other reactive antibodies that see different epitopes.

The reviewer is correct; these ELISAs were performed to confirm the depletion of antibodies specific to the column proteins. They are thus a methodological control, and do not exclude the possibility that antibodies to other epitopes were present in the depleted sera. We have discussed this possibility at length in the discussion, but have now also performed some additional experimental validations, showing that in parasites with Pfs48/45 knock-outs (which also lack Pfs230 – which is bound to the gamete surface by association with Pfs48/45), the sera depleted of Pfs48/45 and Pfs230 antibodies specific to the epitopes of our recombinant proteins recognise proteins on the gamete surface. This provides unequivocal proof that surface proteins other than these two exist and are recognised by naturally occurring antibody responses. We believe these additional experiments satisfy reviewer 1 and 2, who both brought up this issue.

For all individuals the α -Pfs48/45 and α -Pfs230 depleted IgG (containing no antibodies against Pfs48-45 epitopes 1-3, or Pfs230 epitope C, confirmed by ELISA) inhibited transmission to

mosquitoes (table 3). This activity was complement-independent. Importantly, for individuals with sera still available for testing (n=.../6), both the total and depleted IgG recognized unknown antigens on the surface of Pfs48/45 KO gamete (lacking both Pfs48/45 and Pfs230, which binds to the gamete surface via its association with Pfs48/45^{4,20}) (figure 4).

Additionally, in response to this and to some of the points in the reviewers introductory statement, we now provide an additional supplemental table (table S4) detailing the overlap between responses to Pfs48/45, Pfs230 and our novel biomarkers of TRA. This is presented as a heatmap showing z-scores and seropositivity, for all individuals with high-level TRA. We feel this table provides increases the clarity of our data; clearly some responses are highly prevalent in individuals lacking responses to 230 and 48/45.

16. SMFAs were done in duplicate but the assay was done two different ways? Why is there a preferred read out? For consistency, shouldn't the read out be the same? It is not clear how two pools of 10 mosquitoes (using the GFP-luc strain) equivalent to assessing (NF54) oocyst counts in a single individual? Given the variation in the SMFA (as well as DMFA), it is unclear why the authors elected to do only two replicate experiments. There are instances where even 3 replicates are not enough. No technical replicates were performed. The statement that such differences make no difference (referencing their own work) to the integrity of the analyses is debatable. Robustness requires consistency and replication (both technical and biological). This seems to be missing here and borne out in Supplemental Table 6 when triplicate SMFAs were actually performed.

The reviewer is correct that the SMFA was conducted variously by dissection and counting of midgut oocysts (with NF54 strain *P. falciparum* parasites), or with a luminescence read-out, using a luciferase expressing transgenic NF54 strain. The variation in SMFA read-out was not directed to particular samples or duplicates, but reflects a change of practice in the laboratory these SMFAs were conducted brought about in the year the assays were conducted. Luminescence based SMFA increases the assays throughput and provides indistinguishable read-outs in terms of transmission reducing activity (Stone et al. 2015, Vos et al. 2016).

The reviewer is absolutely correct in their assertion that robustness requires technical and biological repetition. Indeed, if we were attempting an analysis based on more granular detail of transmission reducing activity (i.e. non-reducer/low reducer/mid-level reducer/high reducer), then duplication or triplication would be merited. TRA in the SMFA is notably very variable at low levels of TRA, but high level TRA has been shown to be very robust; 90% TRA in particular has been supported as a replicable level of high TRA in previous studies, notably by van der Kolk and colleagues (Parasitology, 2005). Aware of the scale and inherent variability of the assays we required, we chose to perform all assays once, to duplicate within an inclusive window of high TRA (>80%), and to include in our analysis only those samples with the highest, replicable TRA (>90%). At the other end of the scale, we excluded samples from the comparison group that had any indication of partial TRA – only samples with <10% TRA were used as the comparator group in our analysis. The experiments the reviewer mentions in supplemental table 6 are serial dilutions of mAb, which are performed in triplicate as standard (to ascertain IC50's).

17. The authors used 30 kDa filters to remove cytokines or reactive species but no other validation or check was done?

No assays were used to detect the presence of cytokines in the material used for SMFA or column purification. These were unnecessary because all antibody material was purified IgG. Protein G columns were used for IgG purification, which do not bind other antibody classes or cytokines. The 30kDa filters used for antibody concentration (after purification) would likely have acted as a second barrier to their presence, but an unnecessary one. Our mentioning them explicitly in the original manuscript drew unnecessary attention to the possibility of cytokine presence. We have edited this section thus:

The possibility that TRA from the column fractions was due to cytokines or reactive ROx/RNOx species, which in circumstances of inflammatory crisis can inhibit parasite development in mosquitoes^{96,97}, was excluded both by our use of protein G binding columns for IgG purification for all samples, and our use of 30kDa filters for antibody concentration prior to mosquito feeding.

18. Figure 1: Are TRA < 0% indicative of transmission enhancement? If not, what does it represent? Why were the data not parsed out according to age? Instead of intensity, why not quantify antigen specific IgG?

TRA <0% technically represents transmission enhancement relative to the controls. However, the degree of 'enhancement' in our study was not unexpected, and is in fact a common feature of such surveys. This is due for the most part to the inherent variability of the SMFA, which has already been discussed. Especially at low oocyst intensities, small natural variations in the mean oocyst intensities of different mosquito groups naturally leads to a lot of noise around the level of the controls in the test mosquitoes with no active TR antibodies. There is limited evidence that biological transmission enhancement exists for *P. falciparum*. If *P. falciparum* enhancement truly exists as a biological process that is related to (low level) gametocyte antibody concentrations, this would have considerable consequences for our understanding of transmission efficiency and, potentially, the future deployment of transmission blocking vaccines. Analogous to the previous comments of reviewer 1 on the necessity to be cautious in proposing new candidate antigens for TRA, we would like to be very cautious in presenting any data on TRA estimates below 0%. Whilst we believe values less than 0% may represent methodological variation, rather than a biological phenomenon, we have provided more detail about the individuals in this range in the methods without drawing any conclusions on potential biological mechanisms or consequences for transmission efficiency:

For individuals with TRA <0%, median TRA was -41.8% (IQR -80.9 - -16.5-). Maximum enhancement relative to controls was at -472.0%.

As discussed in response to reviewer 3, all analysis of TRA on a continuous scale are performed using log transformed relative intensities; this was done to ensure that TRA was not flattened at zero for statistical analysis.

To more rigorously control the DMFA analysis for the potential effect of variable gametocyte density, we now also perform these analysis only on individuals for whom gametocytaemia was confirmed by microscopy (rather than all individuals with infectiousness data). These analysis remain adjusted for gametocyte density as described here. We feel this change in our analysis adds to the robustness of our conclusions regarding the association of antibody responses and infectivity.

19. Figure 2: Antibody breadth is higher in 0-5 year olds, so these children are less infectious to mosquitoes?

This is an excellent point – yes, children had significantly higher antibody mediated TRA on average. Though this is a logical addition to figure 2, we decided to address this with a panel of descriptive box plots/bee swarm graphs in supplemental figure 1, alongside some related graphs.

20. Figure 3: The peptides corresponding to the respective protein is not given anywhere in the manuscript with reference to ≥ 14 of the 61 to ≤14 of the 61 targets (or peptides? or proteins?). Which 14? Same 14? <10/20 TR peptides? Are they part of the > 14? How do they partition? Titer of anti P48/45 and/or P230 antibodies with each combination of peptide targets? Protein targets? The information could be conveyed more effectively.

In the revised manuscript, data on the association of responses to specific peptides are now given in a separate table (table 2) from one with more detail of the proteins they map to (table (table 4).

For the first analysis (>/<14 peptides seropositive) we aimed simply to compare the infectivity of individuals who were seropositive to a large number of the TR associated proteins, with the infectivity of people responding to fewer. For this, we chose 14 seropositive peptides as the benchmark for high reactivity, because a response to 14 was the 3rd quartile of response breadth (i.e. 25% of all individuals were seropositive to 14 or more peptides). The proteins themselves are therefore different for each individual – we are only interested in the frequency of response to any of the 61 peptides. We have improved our description of this approach, which in the first manuscript was only explained in the figure legend.

The next analysis was specific to identified proteins – those with significantly higher seroprevalence among transmission blockers. This was an attempt to focus the analysis on a highly associated subset, providing identifiable markers of TR immunity. We consider this approach less meaningful for the revised manuscript. In table 4 of the revised manuscript we now identify proteins that could be plausible targets of TR antibodies. We have therefore chosen to replace the aforementioned sub-analysis with one associating natural transmission and response to the identified potential TR targets. In the text, we show that removal of Pfs230 and Pfs48/45 seropositive individuals from this comparison does not affect the associations. The approach itself was again based on frequency of response to any of these peptides – however, the subset is now clearly identified in table 4.

Major/Minor comments to further improve clarity:

- **What are the benchmarks that would lead the community to successfully achieve the following goal? Line 22-24, p. 2, introduction: “Identifying the factors responsible for immune TRA could lead to the development of novel malaria transmission blocking vaccines, and provide useful markers of infectiousness.”**

This was intended as a general statement. As we have now more clearly defined our rationale, we are also now cautious in our statements regarding the usefulness of further study of natural TRA, from a vaccine development point of view. We hope the final paragraph of our discussion makes this clear.

- **What is the relevance of transmission-reducing activity to malaria transmission? Mosquitoes remain infectious if only a single oocyst remains productive.**

This is a very valid point, but not one that we feel the current manuscript can address in detail. The SMFA used unnatural oocyst densities in control mosquitoes and thus reductions in oocyst intensity are most appropriate (as discussed above and demonstrated in manuscripts by Miura and Churcher). When exposure (oocyst intensity) is reduced, reductions in oocyst density mirror reductions in the proportion of infected mosquitoes (Churcher, International Journal of Parasitology, 2012). The SMFA is thus optimised to sensitively screen for high levels of TRA that are likely to lead to substantial reductions in the proportion of infected mosquitoes or even total annulment of onward transmission in natural infections. Our analyses confirm this. High level TRA in the SMFA was strongly associated with a reduced proportion of infected mosquitoes in the DMFA and also associated with a failure to infect any mosquitoes in the DMFA.

We have added some detail to our methods section to better explain and define the ‘high-level’ TRA group we use in our analysis.

≥90% reduction in oocyst/luminescence intensity in test mosquitoes relative to controls was classified as evidence of high-level TRA, which previous data indicate is highly replicable⁹². Samples from all individuals with >80% TRA were tested in duplicate to ensure robustness of the functional phenotype and the mean value of duplicate experiments was used to classify individuals as having high level TRA (>90% average reduction) or no evidence of TRA (<10% average reduction).

- **The Table 1 legend suggests that not all samples have available data, yet the text on p. 3 indicates that malariometric data accompanied all plasma samples? Unclear.**

The text on page 3 was wrong, some individuals indeed had incomplete malariometric data. This has now been indicated.

- **Table 3 requires additional legends to explain column headings and predictions based on what software or source.**

Table 3 has been edited, and the specific column removed in the revised manuscript.

- **Clearly indicate stage V gametocytes, if this is what the authors want to convey. “Mature gametocytes” have not been clearly defined. Additional text is needed to explain the selection of non-stage V gametocyte proteins for inclusion in the array.**

This has been edited throughout the manuscript. Detail has also been added to the results and methods sections to explain the presence of proteins not specific to mature gametocytes.

- **It is difficult to follow the origin of samples and how they partitioned during analysis/experimentation. “Individuals from malaria endemic areas were all asymptomatic when sampled, and were either recruited randomly (n=42), or based on the observation of malaria parasites (n=273) or specifically gametocytes (n=276) in blood smears”. For example, the 42 samples were from which country and what age group? A table summarizing this information would be helpful. Presumably the samples were collected at various periods during a single or several transmission seasons. Since sampling early will likely lead to a different exposure to gametocytes than later in the transmission season, how was this factor controlled in the analyses?**

We have added detail to the methods section describing in detail the sampling and recruitment criteria for each location, as indicated above in response to point 5 by the reviewer.

Our sample selection was focused on studies of infectivity, which had been conducted during or shortly following the rainy season and were almost exclusively done on parasite positive asymptomatic individuals. Individuals contributed only one observation to the current study. More detailed longitudinal sampling in gametocyte carriers is required to directly determine the impact of gametocyte exposure on the dynamics of TRA but was not part of the current study. The variation in sampling time-points between studies is part of the reason that gametocyte density is controlled for in all statistical analyses – this is the best available indication of varying parasite exposure we have, and its frequency/intensity will change with the timing and location of sampling. We also make clear in our manuscript that associations with age, though mentioned, should be viewed with the understanding that this sample set is not a single cohort.

To increase the clarity of these sections, we have added a supplemental figure showing the array responses by age for a single study location (Burkina Faso) – in response to reviewer 3’s similar comment regarding variation with location.

- **Of the 6 TRA sera, only 3 had >10 mL. What sort of bias was introduced into the study here?**

No bias would have been introduced by the availability of large serum volumes from certain individuals. Many of the plasma donors were young children where larger volumes are not routinely collected and sometimes ethically difficult to justify. Even for adults, only a minority had such large samples available. The selection for the current manuscript is thus simply an artefact of different sampling structures. The antibody concentration procedure was performed to demonstrate that functional antibodies may be present at concentrations that are too low to achieve high level TRA. No general conclusions on the epidemiology of such low-level functional responses are drawn and we are therefore comfortable with our cautious interpretation of these specific experiments.

- **“Though our samples were not collected from a random survey of a single population, the prevalence and magnitude of antibody responses to Pfs48/45 (Pfs48/45-10C68) and Pfs230 (230CMB31) increased with age, and this increase was significant after adjustment for gametocyte density (figure 20 1 A & E)”. So the increase was not significant unless the analysis was adjusted for gametocyte density. What is the rationale for adjusting gametocyte density if the breadth of responses and $\geq 90\%$ TRA did not correlate with higher gametocyte density? This was not very clear. Are gametocyte-negative samples (if really negative) with TRA indicative of prior exposure to gametocytes? If so, the rationale (which is hard to grasp for biologists or non-modelers) for adjusting gametocyte density becomes even more unclear.**

Our methods of adjusting for gametocyte density are now explained more clearly in the text. Adjusting for gametocyte density is statistically prudent, and conservative – not adjusting generally gave rise to slightly greater effect sizes and significance. Repeatedly stating that tests were adjusted though is clearly inelegant, and the sentence the reviewer refers to here was a particularly bad example. In answer to the reviewer’s question, as in most cases, the significance of this association was improved when not adjusting for gametocyte density, not *vice versa*. We have removed this section of the text to increase clarity.

The reviewer is absolutely correct that gametocyte negative samples may well be sub-microscopic positives. For this reason, we do not at any point present comparisons of immune responses between gametocyte positives and negatives. This is also part of the reason we were so strict in our sub-selection of samples for our principal comparison of individuals with and without TR immunity. Individuals with TR immunity must be gametocyte exposed, but we needed to compare to microscopy positives to ensure gametocyte exposure among the comparison group.

To warn readers who may be less familiar with malaria diagnostics, we have included the following sentence in the methods:

Parasitological status was defined by microscopy acknowledging that this lacks the sensitivity to detect many low-density infections; i.e. microscopy negative samples were not treated as an unexposed control.

- **In the text, the TRA analyses as described were for either P230 or P48/45, but not both. Why not both?**

We now provide additional data for the association of TRA with response to either or both antigens in supplemental table 1. We have also edited all relevant sections of the results to report associations with response to either or both antigens.

- **What was the rationale for preparing a 9X concentration of IgG?**

9X was simply close to the maximum allowed for with the volumes of sera we had available (1ml was used for the samples at physiological concentration, 10ml for concentrated samples). This allowed for a simple, useful conclusion: that a certain, arbitrary degree of concentration would result in TRA that the antibodies at their physiological concentration could not effect. It enabled us to turn a non-blocker into a blocker, proving the existence of lower concentrations of TR antibodies in non-blockers.

- **The authors discussed the issue of potentially missing antibodies to P48/45 and P230 that would otherwise not bind to their protein fragments (i.e., not presenting the native epitopes). The ELISA used the same proteins though instead of native antigen or the full-length protein. It is therefore not clear if the quality control was stringent enough.**

See response to point 15.

• **How was false discovery calculated before it was used to adjust the analyses?**

The statistical reviewer has commented on our use of the Benjamini-Hochberg FDR correction, which is a standard method in microarray analysis. We have adjusted our language in line with the reviewer's comments, which were unclear in the original manuscript.

• **The success of anti-P48/45 and P230 antibody purification methods (e.g., coupling efficiency) in this manuscript were based on rabbit and rat IgG affinity, which is different from human antibodies. Not sure what the authors want to convey with the inclusion of the supplemental table 2.**

We apologize for this lack of clarity and have removed the table to avoid confusion. We initially included this table simply as a proof of concept for the removal of Pfs48/45 and Pfs230 specific antibodies from serum IgG samples. In these instances, the only malaria protein the immunised animals have been exposed to is the protein in the column. Accordingly, the eluted specific antibodies were highly transmission reducing, while the depleted flow-through's lost some or all of their TRA (the human IgGs did not – as they contain antibodies to transmission-blocking epitopes/proteins other than those used in the animal immunisations). As indicated, we have now removed these data in order to not distract the reader.

• **The exact concentration of the antibodies (following concentration) is not provided. What the exact antigen-specific IgG concentration used in the assays? This would be helpful in understanding what 9X concentration means.**

Corrected OD in the ELISA for each antigen has now been added to the legend of table 3.

• **From supplemental figure 2: “(OR were therefore generated with reference (ref.) to mosquitoes feeding on individuals with responses in the first quartile)”. What is (ref.)?**

This was an error. The odds ratios reported in the figure were from logistic regression with a categorical independent variable – all categories being compared to the base (or reference/ref.). In the final draft, we decided to remove these categorical odds ratios, and simply present the figure as a descriptive plot – we accidentally included the old version of the figure in the supplemental information document, but not the appended figure. The ref. is therefore unnecessary, and has been removed from the figure in the supplemental info.

Minor comments (general):

• **Errors in spelling, missing articles or words and poor sentence construction abound. Please correct.**

The manuscript has been thoroughly edited.

• **Malaria is a disease so “malaria life-stage” is nonsensical. The authors probably mean “malaria parasite” or “malarial parasite”.**

This has been changed throughout the manuscript.

• **Be consistent-- “transmission-blocking” or transmission blocking? “TBI” was used on p. 14. Transmission-blocking immunity? Not defined at first use.**

This has been changed throughout the manuscript.

Reviewer #2 (Remarks to the Author):

Malaria transmission-blocking vaccines (TBVs) have been regarded as one of the essential

strategies towards malaria elimination. However, there have been only a few TBV candidates under development (Pfs25, Pfs48/45, Pfs230). Therefore there is an urgent need for screening to identify novel candidate antigens for timely TBV development. To this end, the Authors determined the transmission-reducing (TR) activity of naturally acquired antibodies from 648 malaria-exposed individuals by *Anopheles stephensi* mosquitoes in the standard membrane-feeding assay (SMFA). Immuno-affinity purified α -Pfs48/45 and α -Pfs230 antibodies from a subset of individuals were able to inhibit transmission to mosquitoes, but IgG depleted of these antibodies continued to reduce transmission. To identify responsible antigens to the TR activities, the samples were probed on a microarray comprising 319 gametocyte proteins. Finally, they found out that the TRA was significantly associated with responses to 41 novel proteins. Additionally, in the direct membrane-feeding assay (DMFA), they found out that *Anopheles gambiae* mosquitoes were significantly less likely to become infected, and at a reduced rate, if feeding on natural gametocyte carriers who were reactive to Pfs48/45, Pfs230, or to separate combinations of newly identified TR associated proteins. I think that the results of this systematic, large-scale and comprehensive study will provide TBV community very important information for developing effective TBVs based on the selected antigen(s). All the works were carefully designed, clearly presented, and the manuscript is well written. I have the following comments for the improvement of this manuscript.

Major comments:

1) Novel gametocyte antigens associated with TR efficacy

The Authors identified an association between antibody responses to 41 newly described proteins and naturally acquired immune TRA (Table 3). I am sure that this is the most exciting data in addition to the Pfs48/45 & Pfs230 (well-known TBV vaccine antigens) results, however there is no validation results presented. The Authors themselves claimed that this is not in the scope of this study (future work). I understand how much more efforts are needed to produce such validation results. Even though I still think it is very important and attractive to the readers of this journal that animal antibody against at least one novel gametocyte antigen out of 41 shows the TR efficacy in SMFA.

We thank the reviewer for their comments – they have been very useful. As described, our original manuscript did not sufficiently crystallise our research objectives. Though we did incorporate some functional validation into our study (for Pfs48/45 & Pfs230, for which we had numerous antigen and antibody resources) this was conducted with the aim of justifying the search for additional biomarkers of TRA. The aim of the array analysis was to identify further TRA correlates to expand and prioritise a pool of targets *for further functional characterisation*, for use in vaccine or biomarker development; we were not clear enough that functional validation was the next step for these proteins, nor that many of them may be biomarkers of TRA only.

This said – we enriched our array to include proteins that would be plausible gamete surface proteins, based on their characteristics. We have now focused more on this with our new table 4, and subsequent analysis. The obvious endpoint of this line of analysis is functional validation of involvement in TRA, as the reviewer indicates. One of the proteins we identified as having significantly TRA associated antibody responses was PfGEST. PfGEST has previously been identified as having a crucial role in gamete egress – an essential event proceeding gamete fertilisation in the mosquito midgut (Talman, *Molecular Microbiology*, 2011). Though PfGEST is likely to be exported through the plasma membrane, it is uncertain whether the protein is retained on the gamete surface in *P. falciparum*, or how it interacts with the erythrocyte membrane to facilitate egress. PfGEST (Talman) showed association with *P. berghei* surfaces 12 minutes after activation, but the protein was also present in the cell supernatant – indicating partial surface association in this species. Previous data with another of our TRA associated hits, Pf11-1, indicate a similar role in egress. Excitingly, mAb specific to Pf11-1 appear to limit the number of gametes proceeding to fertilisation (Feng, *Journal of Experimental Medicine*, 1993). With this in mind, we have analysed a pool of

monoclonal antibodies raised against PfGEST. Surface immunofluorescence assays with WT *P. falciparum* gametes show that PfGEST is not retained on the gamete surface, and SMFAs with PfGEST mAb show no significant, replicable TRA. Though disappointing, these data provide valuable insight into the validity of PfGEST as a potential transmission blocking vaccine candidate, and provide an experimental validation for the concept that proteins may be biomarkers of TRA while not being mechanistically linked with the phenomenon.

As described in our response to previous points, we now provide bioinformatics analysis with the aim of predicting which of our TRA biomarkers may be present on the gametocyte/gamete surface. This sub-selection provides a conservative starting point for future validation of additional candidates for their role in naturally acquired TRA.

2) Affinity purification of antibodies against Pfs48/45 & Pfs230

Results section:

“For all six individuals the α -Pfs48/45 and α -Pfs230 depleted IgG (fraction 5, containing no antibodies against Pfs48-45 epitopes 1-3, or Pfs230 epitope C, confirmed by ELISA) inhibited mosquito transmission. This activity was complement-independent.”

Discussion section:

“It is highly unlikely that α -Pfs48/45 antibodies to epitopes other than epitopes 1-3 would have contributed considerably to TRA. Because the activity of α -Pfs230 antibodies is complement dependent, it is similarly unlikely that the TRA we observed in α -Pfs48/45 and α -Pfs230 antibody depleted IgG fractions is due to antibodies to Pfs230 epitopes outside the region captured by the 230CMB product (aa. 444-730).”

Based on the above description, the Authors emphasized that the completeness of the removal of antibodies against both Pfs48/45 R0-10C and Pfs230CMB. As they know there is no proof that the folding of these cysteine-rich recombinant proteins is identical to the native proteins. Moreover, there is no proof that anti-Pfs230 antibody against outside of Pfs230CMB region still remained after the affinity purification by the recombinant proteins. To clarify this uncertainty, I strongly recommend the Authors to measure anti-Pfs48/45 and anti-Pfs230 antibodies in six total IgG samples after the affinity purification (Table 2) by the ELISA using *P. falciparum* gamete extract and monoclonal antibodies. Because the Authors are the establishers of this excellent ELISA system. (Ref. Bousema, T. et al. The Dynamics of Naturally Acquired Immune Responses to Plasmodium falciparum Sexual Stage Antigens Pfs230 & Pfs48/45 in a Low Endemic Area in Tanzania. PLoS ONE 5, e14114 (2010); Ouedraogo, et al. Naturally Acquired Immune Responses to Plasmodium falciparum Sexual Stage Antigens Pfs48/45 and Pfs230 in an Area of Seasonal Transmission. Infect Immun 79, 4957-4964 (2011).

The reviewer is correct that antibodies specific to epitopes of the Pfs48/45 and Pfs230 proteins other than those present in the column bound proteins probably exist in the depleted sera. Though we devoted a significant portion of the discussion to this, we have made clearer in the revised discussion that we fully anticipate their presence, and that we do not think they can explain the consistent, high level TRA observed in the depleted sera SMFAs:

Our affinity purification experiments were based on the binding of antibodies to epitopes present in the Pfs48/45-10C and 230-CMB vaccine constructs, since the proteins' complex conformational structures so far preclude their production as full-length proteins. We thus captured the most potent transmission-reducing antibodies (Pfs48/45 epitopes 1-3, Pfs230 epitopes within the C region)^{15,29,31,53,84} but not necessarily all antibodies to these proteins. Monoclonal antibodies against epitope 5, which would not bind to the Pfs48/45-10C protein, have been associated with weak TR activity⁸⁶. However, given their lower efficacy and the immuno-dominance of epitopes 1-3 in naturally immune sera¹⁵ it is highly unlikely that α -Pfs48/45 antibodies to epitopes other than epitopes 1-3 would have contributed considerably to TRA. This is also supported by the finding that antibodies to epitope 5 are only observed in the presence of antibodies to epitopes 1-3¹⁵. Crucially, IgG depleted of antibodies to Pfs48/45

epitopes 1-3 recognized surface antigens of a knockout parasite line that fails to express Pfs48/45, formally demonstrating recognition of other target antigens by depleted IgG. Because all known α -Pfs230 antibodies are complement dependent^{40,43}, whilst TRA in the antibody-depleted fraction is complement independent, a role for residual α -Pfs230 antibodies (responsive to epitopes outside the aa. 444-730 region of Pfs230 represented by the 230CMB product) is similarly unlikely. Our findings of TRA in the absence of measurable α -Pfs48/45 or α -Pfs230 antibodies, together with persisting gamete recognition and functional TRA after depleting IgG of antibodies to these two targets, thus strongly suggest that responses to other antigens may contribute to naturally acquired TRA.

Because antibodies to other epitopes of Pfs48/45 and Pfs230 are likely to be present but unlikely to be functionally responsible for the high level of TRA observed after antibody depletion, we consider the proposed experiments with native protein of limited value. We would argue that proving the presence of these antibodies doesn't achieve much (it does not prove their functionality). Instead, we believe we can move a step beyond this by showing that antibodies other than those specific to Pfs48/45 and Pfs230 bind proteins on the gamete surface. The presence of antibodies to other epitopes of Pfs48/45 and Pfs230 is expected, the presence of other plausible surface proteins is thus far entirely hypothetical.

To this end, we have performed IFAs using gametes lacking the Pfs48/45 or Pfs230 proteins (Pfs48/45 KO parasites, from van Dijk 2001), proving that the depleted sera contain antibodies other than those specific to these proteins that bind the surface of gametes. Proving their functionality is outside the scope of this manuscript, however, we feel these small experiments provide a more stable basis for the search for TRA biomarkers than was present in the original manuscript.

It is important to re-iterate that the antibody depletion experiments provide one of several threads of evidence that responses to other antigens are involved but by no means is the only evidence for targets other than Pfs48/45 and Pfs230. In the above table (page 12) we summarize findings from individuals with high-level TRA in relation to ELISA results. The used fragments of Pfs48/45 and Pfs230 are immune-dominant, as highlighted in the discussion, and it is therefore highly unlikely that antibody responses to alternative epitopes of these proteins are responsible for TRA in the absence of measurable responses to the dominant epitopes.

Minor comments:

1) Results section:

“Thirty-four of the 42 proteins....Six were primarily expressed by asexual stages; one by sporozoites; two by immature gametocytes; 12 by gametocytes/gametes; and 21 were shared more equally between multiple life stages.....”

Please add this information in the current Table 3. This modification will be helpful to understand easily.

In response to a point by reviewer 1 (point 11) we decided to remove this broad categorisation in favour of a more detailed post-hoc analysis of gametocyte specificity, based on gene expression data from 11 proteomic databases. This has replaced the life stage specific classification in what is now table 4. We also provide more detailed information in table 4 on predicted cellular location and function. As indicated in response to reviewer 11, if specific life stage classifications are considered valuable in addition to the analysis of gametocyte expression, we can include these in a future revision.

2) Figure 1 D&H

Please add (Figure 1 D&H) in the text.

This has been changed.

3) Table 2

“Age/Sex” should be “Sex/Age”.

This has been changed.

4) Page 12, Line 3

“longer than 15 tears” should be “longer than 15 years”.

This has been changed.

Reviewer #3 (Remarks to the Author):

I will restrict my comments to the statistical analyses. These analyses are mostly well done but are not always entirely explained.

I have some questions about Figures 1A and 1E. The p-values are said to be "from logistic regression", but this doesn't tell us what hypothesis has being tested or what specific test has being used. Has a statistical test been done to detect differences in average seroprevalence between any of the three age groups (that would be a test on 2 df)? Has a likelihood ratio test been used perhaps? Analogous questions could be asked about Figures 1B and 1F where the p-values are from linear regression.

The p values for figures 1A and E come from a likelihood ratio test derived from logistic regression, adjusted for gametocyte density. The p values for figures 1B and F come from an F test, derived from linear regression, adjusted for gametocyte density. The reviewer is correct in their assumption that the single p value in each instance represents the significance of differences in seroprevalence or normalised OD between any of the three age groups. These details have been clarified in the text and figure legends.

What do the error bars represent in Figures 1A and 1E? Are they standard errors? Or standard deviations? Or CIs? Do the error bars account for gametocyte density? What are the sample sizes in the three age groups?

The error bars represent exact binomial 95% CIs. Adjustment for gametocyte density has been performed for all relevant statistical analyses, but not for error bar calculation.

I notice that the study participants from The Netherlands are all 26 yrs or older, whereas the subjects from Gambia and Cameroon are all young. Hence the apparent age effects in Figures 1A-B, 1E-F are confounded with possible geographic differences between The Netherlands vs Gambia and Cameroon. Have the authors checked that the apparent age effects are not just differences between populations?

This is a valid concern. As indicated in table 1, the age range of the participants from different locations do vary, based on differing recruitment criteria. Something that was perhaps unclear though is that in the graphs of age and TRA, only individuals from endemic areas were assessed. This was exactly because of those stark differences in age and because Dutch individuals obviously had markedly different exposure histories – so are inappropriate in an analysis (albeit broad) of age and immunity.

Though we think this should address the reviewers concern, we have added a supplemental version of figure 1 using data from Burkina Faso only (which has significant sample numbers in all age brackets) – showing that the same patterns are present. That is, that breadth and magnitude are higher in children <5 than in older age groups.

The linear relationships in Figures 1D and 1H don't look very convincing. This seems partly because of the scale used to measure TRA. I am guessing that TRA is measured as $\max(0, (O_c - O_t) / O_c)$ where

O_t and O_c are the oocyst numbers in test and control mosquitoes respectively. This measurement scale has the problem of compression of values around zero. Would an alternative, less constrained, TR measure such as log(O_t/O_c) be more suitable for linear correlation analyses?

The reviewer is correct that we had flattened the TRA presented on the Y axis at 0%. We agree that conducting linear correlation analysis on data that is in effect artificially compressed is imperfect. Based on the reviewers comments we re-processed this data using the log(O_t/O_c) method, and used this data for all statistical analyses.

As this graph was commented on by reviewer 1 also, and as it was of borderline interest in our initial submission, we have decided to remove it. In the results section and supplemental information figure 1, we still describe the association of some factors with TRA on a continuous scale. In all these cases, we now use the log(O_t/O_c) method. For presentation (figure S1) we retain the linear TRA data for presentation only – all p values are derived from log relative infectivity log(O_t/O_c), as described in the figure legend and data analysis section of the methods:

Figure S1... TRA on a continuous scale (B & C) is presented as absolute TRA for clarity, but the p values are from simple linear regression using log relative infectivity (log[oocysts in treatment mosquitoes/oocysts in control mosquitoes]), to normalise the data and avoid compression at zero.

The manuscript refers to the p-value adjustment as "adjustment for false discovery", which doesn't quite make sense. A more correct way to describe this process would be say that "p-values were adjusted by the Benjamini and Hochberg method to control the false discovery rate below 5%". The adjusted p-values can be referred to as "FDR" values or as "q-values".

We appreciate the reviewers comment, and have edited all reference to FDR adjustments as per their suggestion.

Rather than simply saying that statistical analysis was conducted using R, it would be more usual to cite the major R packages that were used. For the microarray analysis, it seems that the limma package as been used.

Appropriate references have been added.

The microarray analysis seems basically sound, but the background correction step, with negative values converted to 1, is rather crude. The limma package provides automatic processing of GenePix files and much better background correction methods than used here. The normexp background correction method is one that might well be tried.

We thank the reviewer for pointing this out to us, and have now adopted the backgroundcorrect function (with type="normexp", and offset=50) in our data processing. This has affected the all subsequent analysis subtly – so all statistical values and figures have been changed in the text.

Reviewers' comments:

Reviewer #1 (Remarks to the Author):

The response regarding the limited utility of “dose-dependent response data” in their assays is interesting, since this is a classical hallmark of understanding antibody specificity! However, the authors have provided additional analyses and have addressed to the best of their abilities the comments revolving around this criticism, without having to do 2-3 more years of work.

“713 Data availability

714 The data supporting the findings of the current study are available from the corresponding author

715 on reasonable request.”

The above is unacceptable. Please make sure the MS data is uploaded to a public repository, as this is expected of the proteomics community by international MS standards and their proteomics expert should have proposed to do this already in the first place. The above section should include, for example, an accession number to the project data repository upload. It really is quite distressing that this was not part of the entire manuscript submission process in the first place. The lead authors should refer independently to the MIAPE guidelines document and note that there are several public repositories available, including PRIDE, Peptide Atlas, GPMDB, MassIVE, etc. Please also refer to: <https://doi.org/10.1093/nar/gkw936> if you need guidance from other mass spectrometry experts. It is appreciated that the authors understood that a careful comparison of stage-specific proteomes is needed (albeit in the future), as PlasmoDB has a lot of proteomic data that have not been rechecked since they were first published. In addition, several of the projects only use a simplified bioinformatics approach, regardless if the approach is the correct one to use by itself.

Overall, the authors should indeed be congratulated and this reviewer is grateful to them for spending time and effort considering carefully the comments from the initial submission. The changes support the publication of the manuscript (once the MS data has been confirmed to be uploaded and the repository link/number is provided to the journal staff). This the only minor fix that is required before publication.

Reviewer #2 (Remarks to the Author):

The Authors have answered to all of my comments appropriately, added further experiments used Pfs48/45 KO parasites, and finally revised the manuscript with significant improvement and clarity.

Only one correction required:

Line 590: "To ensure that that the" should be "To ensure that the"

Reviewer #3 (Remarks to the Author):

The authors responded to my comments in a helpful fashion, but the following issues still remain. The statistical methods are still not fully described.

There is still no explanation in the manuscript of what the error bars mean in Figures 1A or 1B. If the exact binomial option in STATA has been used, then the bars should be described as Clopper-Pearson confidence intervals. Same comments for Supp Figure 6 A&B. Note that the Nature Communications submission guidelines and the Nature Communications editorial reporting checklist both explicitly require that all error bars be defined.

The Figure 3A legend needs to explain what sort of CIs these are (e.g., Clopper-Pearson).

The Figure 3B legend describes a non-standard definition for the box plots, with the whiskers as 5% and 95% quantiles. Does this mean that the top and bottom 5% of values have been suppressed on the plot (no outliers are shown)? Is there any reason why the standard Tukey definition has not been used? What about all the other box plots in the manuscript for which the whiskers are not defined? Is the construction of the other box plots the same as for Figure 3B or different?

Ritchie et al (Nucleic Acids Research 43(7):e47, 2015, Pubmed 25605792) would be a much more up-to-date reference for limma than the 2005 book chapter currently cited.

The authors say in their rebuttal that they now use normexp background correction for the microarray data. That seems good, but there is no mention of this in the manuscript. Quite the opposite, the manuscript says that local background subtraction has been used.

General changes

Changes have been made throughout the manuscript in order to comply with Nature communications manuscript checklist for formatting. Major changes to wording have been made to the abstract, and final paragraph of the introduction, so that these sections are presented in the present tense.

Response to reviewers' comments:

Reviewer #1 (Remarks to the Author):

The response regarding the limited utility of “dose-dependent response data” in their assays is interesting, since this is a classical hallmark of understanding antibody specificity! However, the authors have provided additional analyses and have addressed to the best of their abilities the comments revolving around this criticism, without having to do 2-3 more years of work.

We agree that dose-dependent response data would provide valuable insight into antibody specificity, and appreciate the reviewers understanding that detailed investigation of this was outside the scope of the current study.

“713 Data availability

714 The data supporting the findings of the current study are available from the corresponding author

715 on reasonable request.”

The above is unacceptable. Please make sure the MS data is uploaded to a public repository, as this is expected of the proteomics community by international MS standards and their proteomics expert should have proposed to do this already in the first place. The above section should include, for example, an accession number to the project data repository upload. It really is quite distressing that this was not part of the entire manuscript submission process in the first place. The lead authors should refer independently to the MIAPE guidelines document and note that there are several public repositories available, including PRIDE, Peptide Atlas, GPMDB, MassIVE, etc. Please also refer to: <https://doi.org/10.1093/nar/gkw936> if you need guidance from other mass spectrometry experts. It is appreciated that the authors understood that a careful comparison of stage-specific proteomes is needed (albeit in the future), as PlasmoDB has a lot of proteomic data that have not been rechecked since they were first published. In addition, several of the projects only use a simplified bioinformatics approach, regardless if the approach is the correct one to use by itself.

Overall, the authors should indeed be congratulated and this reviewer is grateful to them for spending time and effort considering carefully the comments from the initial submission. The changes support the publication of the manuscript (once the MS data has been confirmed to be uploaded and the repository link/number is provided to the journal staff). This the only minor fix that is required before publication.

In response to the primary comment regarding data availability by reviewer 1, the novel *P. falciparum* Schizont mass spectrometry proteomics data have been deposited to the ProteomeXchange Consortium (<http://proteomecentral.proteomexchange.org>) via the PRIDE partner repository with the dataset identifier PXD008250. The project name is: "*Plasmodium falciparum* schizont proteome". The username/password for access to the data are: reviewer29159@ebi.ac.uk/Wj74vqM1. The data will be made available without password protection if final approval of the manuscript is given. All other mass spectrometry data are already in the public domain, are referenced specifically in the manuscript, and are available through the PlasmoDB depository.

Changes to the manuscript are in the data availability section, which now reads:

'The data supporting the findings of the current study are either provided in the supplementary materials, or have been deposited in the DRYAD data depository under accession number ... The novel *P. falciparum* Schizont mass spectrometry proteomics data have been deposited to the ProteomeXchange Consortium (<http://proteomecentral.proteomexchange.org>) via the PRIDE partner repository¹¹⁴ with the dataset identifier PXD008250.'

Additional data to be uploaded to dryad have been detailed in the cover letter.

Reviewer #2 (Remarks to the Author):

The Authors have answered to all of my comments appropriately, added further experiments used Pfs48/45 KO parasites, and finally revised the manuscript with significant improvement and clarity.

Only one correction required:

Line 590: "To ensure that that the" should be "To ensure that the"

This typo has been corrected.

Reviewer #3 (Remarks to the Author):

The authors responded to my comments in a helpful fashion, but the following issues still remain. The statistical methods are still not fully described.

There is still no explanation in the manuscript of what the error bars mean in Figures 1A or 1B. If the exact binomial option in STATA has been used, then the bars should be described as Clopper-Pearson confidence intervals. Same comments for Supp Figure 6 A&B. Note that the Nature Communications submission guidelines and the Nature Communications editorial reporting checklist both explicitly require that all error bars be defined.

The Figure 3A legend needs to explain what sort of CIs these are (e.g., Clopper-Pearson).

The reviewer is correct that the error bars are Clopper-Pearson CIs. All error bars are now fully explained as described in the manuscript with tracked changes.

The Figure 3B legend describes a non-standard definition for the box plots, with the whiskers as 5% and 95% quantiles. Does this mean that the top and bottom 5% of values have been suppressed on the plot (no outliers are shown)? Is there any reason why the standard Tukey definition has not been used? What about all the other box plots in the manuscript for which the whiskers are not defined? Is the construction of the other box plots the same as for Figure 3B or different?

This was an error – the box plots are standard Tukey plots. We have now described this explicitly in the data analysis section, corrected our figure legends as indicated, and have added box plot outliers to figure 3 to avoid confusion. We had originally removed the outliers from these plots because we show all data points in different colours as bee swarms, making the inclusion of outliers of related (identifiably linked) colours difficult. We now include the outliers as hollow black circles.

Ritchie et al (Nucleic Acids Research 43(7):e47, 2015, Pubmed 25605792) would be a much more up-to-date reference for limma than the 2005 book chapter currently cited.

This reference has been updated.

The authors say in their rebuttal that they now use normexp background correction for the microarray data. That seems good, but there is no mention of this in the manuscript. Quite the opposite, the manuscript says that local background subtraction has been used.

This was an error. The normexp background correct function was used for all local background adjustment. Our language has been changed as indicated in the manuscript.